# An ultra-conserved poison exon in the *Tra2b* gene encoding a splicing activator is essential for male fertility and meiotic cell division

Caroline Dalgliesh [1,2], Saad Aldalaqan[1,2], Christian Atallah[3], Andrew Best[1], Emma Scott[1,2], Ingrid Ehrmann[1,2], George Merces[4,5], Joel Mannion[1], Barbora Badurova[1], Raveen Sandher[1], Ylva Illing[6], Brunhilde Wirth [6,7,8], Sara Wells[9], Gemma Codner[9], Lydia Teboul [9], Graham R Smith [3], Ann Hedley [3], Mary Herbert [1], Dirk G de Rooij [10,11], Colin Miles[12], Louise N Reynard[1] & David J Elliott [1,2✉]

## Abstract

**The cellular concentrations of splicing factors (SFs) are critical for controlling alternative splicing. Most serine and arginine-enriched (SR) protein SFs regulate their own concentration via a homeostatic feedback mechanism that involves regulation of inclusion of non-coding 'poison exons' (PEs) that target transcripts for nonsense-mediated decay. The importance of SR protein PE splicing during animal development is largely unknown despite PE ultra-conservation across animal genomes. To address this, we used mouse genetics to disrupt an ultra-conserved PE in the *Tra2b* gene encoding the SR protein Tra2β. Focussing on germ cell development, we found that *Tra2b* PE deletion causes azoospermia due to catastrophic cell death during meiotic prophase. Failure to proceed through meiosis was associated with increased *Tra2b* expression sufficient to drive aberrant Tra2β protein hyper-responsive splice patterns. Although critical for meiotic prophase, *Tra2b* PE deletion spared earlier mitotically active germ cells, even though these still required *Tra2b* gene function. Our data indicate that PE splicing control prevents the accumulation of toxic levels of Tra2β protein that are incompatible with meiotic prophase. This unexpected connection with male fertility helps explain *Tra2b* PE ultra-conservation and indicates the importance of evaluating PE function in animal models.**

**Keywords** Poison Exon; Alternative Splicing; Ultraconserved Genome Sequence; Spermatogenesis; Fertility
**Subject Categories** Development; RNA Biology

## Introduction

Almost all human genes are split into exons and introns, and thus depend on splicing to join exons together and make functional mRNAs (Berget et al, 1977). However, not all exons are protein coding (Aspden et al, 2023). In particular, a group of non-coding exons called "poison exons" (abbreviated PEs) introduce premature stop codons into mRNAs. PEs block mRNA translation and target mRNAs for degradation by nonsense-mediated decay (NMD). PEs have been found in over a third of human protein-coding genes in kidney, liver, CNS and eye mRNAseq datasets (Lim et al, 2020). Despite this, PEs are notoriously elusive, under-represented in mRNAseq datasets because of degradation via NMD (Kovalak et al, 2021), and commonly not detected by genome annotation algorithms. Poison exons have been identified as potential therapeutic targets in cancer and neurological disease (Leclair et al, 2020; Neumann et al, 2020; Preußner et al, 2023; Naro et al, 2019; Lin et al, 2023), so it is important to understand their normal functions in the healthy body.

A particularly intriguing group of PEs are embedded in the genes encoding serine/arginine-enriched (SR) RNA-binding proteins (Ni et al, 2007; Lareau et al, 2007). These SR protein gene PEs are potentially very important since they are encoded by "ultra-conserved" genome sequences. Ultra-conserved sequences are defined as being identical between human and mouse, and are

[1]Newcastle University Biosciences Institute (NUBI), Central Parkway, Newcastle University, NE1 3BZ Newcastle upon Tyne, UK. [2]Newcastle University Centre for Cancer, Newcastle University Institute of Biosciences, Newcastle NE1 3BZ, UK. [3]Bioinformatics Support Unit, Faculty of Medical Sciences, Newcastle University, Newcastle upon Tyne NE2 4HH, UK. [4]Newcastle University Biosciences Institute (NUBI), Innovation, Methodology and Application (IMA) Research Theme, Faculty of Medical Sciences, Newcastle University, NE2 4HH Newcastle upon Tyne, UK. [5]Image Analysis Unit, Faculty of Medical Sciences, Newcastle University, Newcastle upon Tyne, UK. [6]University of Cologne, Institute of Human Genetics, Kerpener Str. 34, 50931 Cologne, Germany. [7]Center for Molecular Genetics, University of Cologne, Cologne, Germany. [8]Center for Rare Diseases Cologne, University of Cologne, Cologne, Germany. [9]The Mary Lyon Centre at MRC Harwell, Harwell Campus, Oxfordshire OX11 0RD, UK. [10]Reproductive Biology Group, Division of Developmental Biology, Department of Biology, Faculty of Science, Utrecht University, Utrecht, The Netherlands. [11]Center for Reproductive Medicine, Academic Medical Center, University of Amsterdam, Amsterdam, the Netherlands. [12]Newcastle University Translational and Clinical Research Institute, Newcastle NE1 3BZ, UK.
✉E-mail: david.elliott@ncl.ac.uk

also almost identical across more divergent vertebrate species including rat, chicken and dog (Snetkova et al, 2022). SR protein gene PEs operate in feedback loops: increasing levels of SR protein expression activate PE inclusion into their own pre-mRNAs, and thereby block further SR protein production. In this way, SR protein PEs have been proposed to maintain robust transcriptomes by preventing "spikes" occurring in SR protein expression levels (Jangi and Sharp, 2014). Recent landmark studies (Thomas et al, 2020; Leclair et al, 2020) using cultured cell lines have shown SR protein PEs are not essential for mitotic proliferation, and rather have tumour suppressor activities by acting as a brake on SR protein expression. However, whether normal tissues and developmental stages need to be protected from spikes in SR protein expression, and if so why, is almost completely unknown. This knowledge gap within living animals is particularly salient since (somewhat surprisingly), genetic deletion from mice of non-coding ultra-conserved sequences that operate as transcriptional enhancers cause only subtle developmental effects insufficient to compromise survival and/or fertility (Snetkova et al, 2022).

Here we have sought to address the developmental importance of a PE embedded within the SR protein gene *Tra2b* (Fig. 1A). Increased Tra2β protein auto-activates *Tra2b* PE splice inclusion, thus blocking further Tra2β protein production (Stoilov et al, 2004), and also cross regulates a PE in the *Tra2a* gene (a paralog of *Tra2b* which encodes the similar Tra2α protein) (Best et al, 2014b; Leclair et al, 2020; Grellscheid et al, 2011a). The *Tra2b* PE is 100% identical between human, mouse and rat, and 96% identical across 300 million years of evolution between human, chicken and lizard. This level of *Tra2b* PE ultra-conservation even exceeds some *Tra2b* protein coding exons: for example, mouse *Tra2b* exon 4 is only 93.65% identical with human. Despite its extreme conservation, deletion of the human *TRA2B* PE does not prevent mitotic division in cultured cells, but rather promotes cellular proliferation (Thomas et al, 2020; Leclair et al, 2020). Similarly, transition between transplanted naïve and effector T cells in response to antigen stimulation does not take place properly without the *Tra2b* PE, suggesting that skipping of this exon helps drive mitotic cell proliferation, similar to the model of PE function in cancer cells (Karginov et al, 2024).

Intriguingly, particularly high levels of *Tra2b* PE inclusion are detected in adult mouse testis (Grellscheid et al, 2011a), suggesting a possible role in spermatogenesis. Importantly, spermatogenesis involves specialised cell types including both mitotically proliferating cells and cells undergoing meiosis (Fig. 1B) that cannot be investigated using cell culture approaches used to globally monitor PE function (Thomas et al, 2020; Leclair et al, 2020). Both splicing factor expression and alternative splicing are dynamically modulated during spermatogenesis and important for male fertility (Naro et al, 2021; Schmid et al, 2013; Oud et al, 2022; Rong et al, 2023). However, it has been unknown whether the *Tra2b* poison exon is important during spermatogenesis. Here, we find that the *Tra2b* PE is essential during meiotic prophase to prevent hyper-activation of particularly sensitive splice sites with high associated levels of Tra2β binding, but not required by mitotically proliferating cells. This discovery of exquisite cellular specificity for PE function provides a paradigm to understand ultra-conserved SR family PE function during the development of complex animal tissues. These data indicate that PEs might have unexpected roles in other developmental programmes, and are potentially important for the development of molecular therapies that target PE function (Leclair et al, 2020).

# Results

## SR protein family PEs are frequently spliced within the testis

We first carried out isoform-specific PCR analyses to monitor SR family PE inclusion across different mouse tissues (Fig. 1C). As shown previously, particularly high levels of *Tra2b* PE inclusion were detected in the testis (Grellscheid et al, 2011a). PE-containing splice isoforms for *Tra2a*, *Srsf3*, *Srsf4*, *Srsf7*, *Srsf9*, *Srsf10* and *Srsf11* mRNAs were also detected within the testis in addition to variable expression across other tissues. The only exceptions were *Srsf5* and *Srsf6*, for which PE inclusion was generally low or not detected (Fig. 1C). Alignment of publicly available mRNAseq data (Data ref: (Soumillon et al, 2013)) also detected high *Tra2b* poison exon inclusion within the adult testis compared to liver and brain (Appendix Fig. S1A). Aligned mRNAseq data from purified adult testis cell types (Data ref: (Soumillon et al, 2013)) showed higher levels of *Tra2b* poison exon containing isoforms within spermatocytes and spermatids compared to spermatogonia and Sertoli cells (Appendix Fig. S1B). Thus the *Tra2b* PE is upregulated in meiotic cells.

## The *Tra2b* PE is essential for male fertility

The high levels of SR protein gene PE-containing isoforms detected within the testis did not necessarily imply functional importance, since these isoforms could result from meiotic changes to the NMD environment that may generally stabilise frameshifted transcripts (Shum et al, 2016; Jones and Wilkinson, 2017). To thus test whether SR protein gene PE splicing is important within the testis, we deleted the *Tra2b* PE within the male germ line and monitored the effect on germ cell development.

CRISPR/Cas9 was used to flank the mouse *Tra2b* PE with *LoxP* sites (Fig. 1A and Methods) and create a new conditional *Tra2bPE*$^{fl}$ allele. Homozygous deletion of the *Tra2b* PE was then driven within germ cells using the germ cell-specific *Vasa-Cre* transgene (Gallardo et al, 2007). *Vasa-Cre* produces a recombinase that is specifically active in embryonic germ cells of both male and female mice from E15 (embryonic day 15, Fig. 1B). *Vasa-Cre*-mediated recombination between *LoxP* sites is reported to be 97% complete by birth (Gallardo et al, 2007). Experimental mice were generated with the following genotypes: *Tra2bPE*$^{fl/+}$ (hereafter known as wild type); *Tra2b*$^{fl/-}$ and *Tra2b*$^{fl/+}$;*Vasa-Cre* (hereafter known as *Tra2b-PEhet*); and *Tra2b*$^{fl/-}$;*Vasa-Cre* (hereafter known as *Tra2b-cPEko*). Details of the crosses used to generate these experimental mice are given in Fig. EV1A–C. Note that since *Vasa-Cre* is only expressed in germ cells, such *Cre*-mediated recombination removes the *Tra2b* poison exon specifically within germ cells (cells in the developmental pathway leading to sperm). This means that somatic cells within these *Tra2b-cPEko* testes (Sertoli cells, Leydig cells and myoid cells) will still have the *Tra2b* poison exon.

Initial analyses of these mice showed that the *Tra2b* PE is essential for male fertility. The testes from adult *Tra2b-cPEko* mice were significantly smaller compared with wild type mouse testes (Fig. 1D,E). Adult male *Tra2b-cPEko* mice also failed to produce any litters when kept in cages with female wild type mice, while crosses of wild type littermates consistently produced pups (Fig. 1F). Interestingly, crosses of adult female *Tra2b-cPEko* mice with wild type male mice also failed to produce litters (Fig. EV1D). Hence the

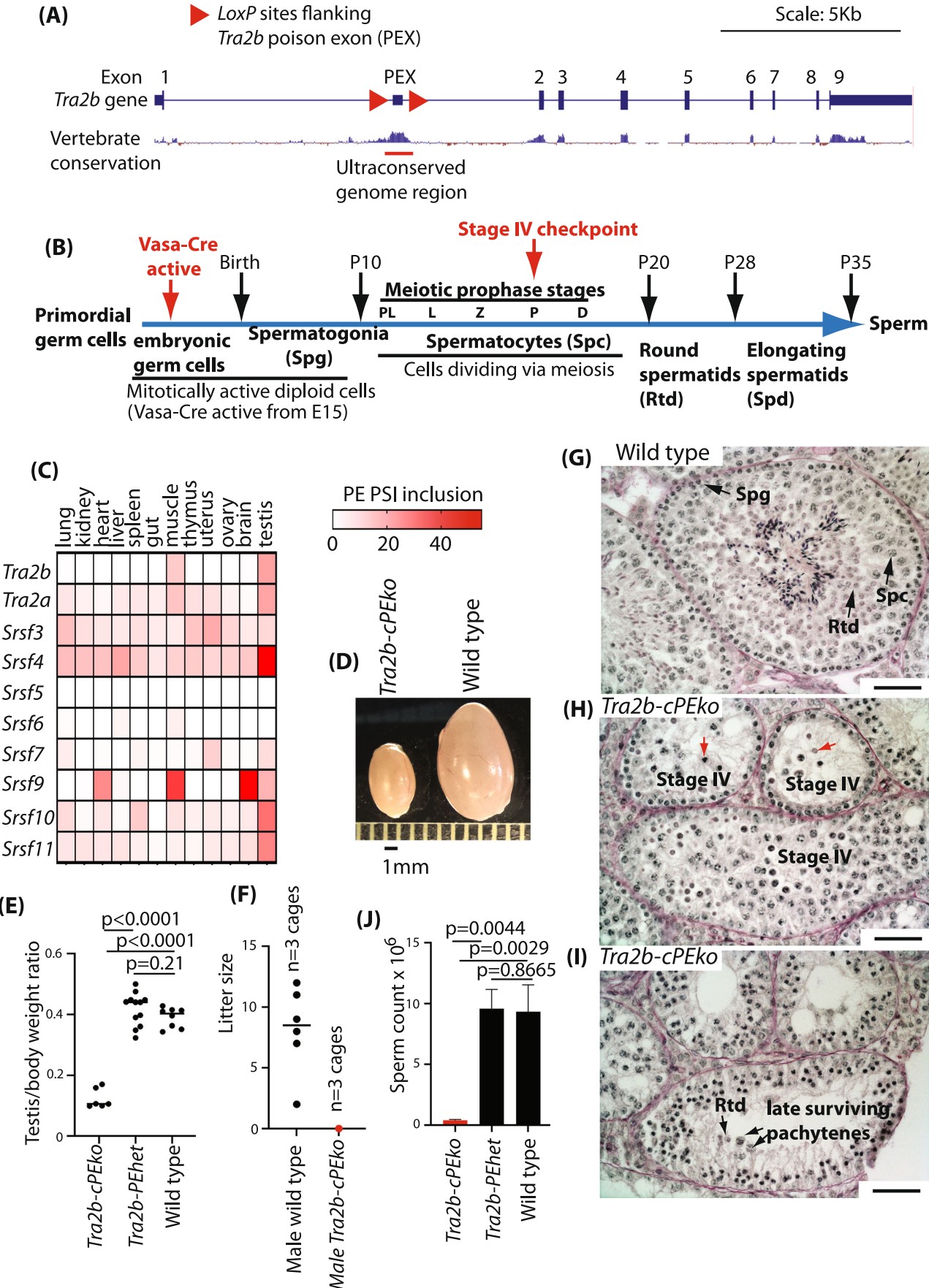

Figure 1.   The *Tra2b* PE has an essential role in meiosis required for male fertility.

(A) Screenshot from the UCSC genome browser (http://genome.ucsc.edu) (Karolchik et al, 2014) showing the mouse *Tra2b* gene, with PhyloP representation of base-wise vertebrate genome conservation, and the positions of the LoxP sites used to make the conditional *Tra2b-cPEko* mouse line. (B) Schematic of germ cell development showing the cell types between primordial germ cells and sperm. The timeframe of cell type appearance in embryonic development from E15 and during the first wave of neonatal spermatogenesis is shown above (postnatal days P1–P35). (C) Anatomic levels of inclusion of mouse SR protein gene PEs. Percentage Splicing Inclusion (PSI) levels were monitored using RT-PCR amplification between primers in exons flanking the PE for each gene. Mean data are shown from $n = 5$ adult mice (8 week old, tissues from 3 females and 2 males plus one extra testis sample). (D) Testis morphologies of wild type and *Tra2b-cPEko* 19-week-old adult testes (scale shown in millimetres). (E) Testis:body weight ratios of different genotype adult mice (age range 9–25 weeks). Sample sizes $n = 6$ testes from *Tra2b-cPEko*, $n = 11$ testes from *Tra2b-cPEhet* mice and $n = 8$ testes from wild type mice. The mean is shown as a horizontal black line, and *P* values were calculated using a t test. (F) Litter sizes of *Tra2b-cPEko* males and wild type males after crossing with wild type female mice. Individual litter sizes from wild type mice are shown as black dots, and the mean as a horizontal line. No litters were obtained from *Tra2b-cPEko* mice (red dot). 3 breeding cages of each cross were maintained until each of the wild type cages had produced a litter (male mice mated aged 7–11 weeks). (G) Micrograph of wild type adult testis section stained with periodic acid Schiff (PAS). Slides analysed from $n = 1$ wild type testes. Black arrows point to examples of spermatogonia (abbreviated Spg), spermatocytes (abbreviated Spc) and round spermatids (abbreviated Rtd). Scale bar $= 20\ \mu m$. (H, I) Micrographs of PAS-stained adult *Tra2b-cPEko* testis sections. Panel (H) shows 3 stage IV arrested tubules: red arrows point to apoptotic pachytene cells detected by PAS staining. Panel (I) shows rare tubule containing late surviving pachytene cells and some round spermatids. Scale bar $= 20\ \mu m$. Abbreviations for cell types as in (B). Slides analysed from $n = 2$ *Tra2b-cPEko* testes. Scale bar $= 20\ \mu m$. (J) Sperm counts from wild type and *Tra2b-cPEko* mice. Sample size $n = 2$ *Tra2b-cPEko*; $n = 3$ *Tra2b-cPEhet* mice; $n = 5$ wild type mice (age range 18–25 weeks). Graph shows mean $+/-$ SD. Statistical significances were measured using unpaired t tests (Graphpad prism). Source data are available online for this figure.

*Tra2b* PE is also required for female fertility (this is not discussed further in this report).

## The *Tra2b* PE is essential for meiotic cell division

Histological examination revealed that wild type adult testis contained all the expected germ cells at different stages of spermatogenesis, while there was an almost complete absence of haploid spermatids in the testes from *Tra2b-cPEko* littermates (Fig. 1G–I). This indicated the small testis size and infertility of *Tra2b-cPEko* mice was caused by a block in germ cell development during meiosis. More precise developmental staging of testis sections from adult *Tra2b-cPEko* mice revealed this block to be at Stage IV of spermatogenesis (Fig. 1B,H) (de Rooij and de Boer, 2003). The most advanced type of germ cells in an epithelial stage IV tubule in *Tra2b-cPEko* mouse testes are spermatocytes in the pachytene stage of the meiotic prophase. In these tubules, apoptosis of pachytene spermatocytes could be observed (indicated by red arrows in Fig. 1H).

A small number of later-stage pachytene cells and round spermatids were occasionally detected in testis sections from adult *Tra2b-cPEko* mice, indicating low level leakage of this Stage IV arrest phenotype (Fig. 1I). There was also a low level of sperm detected within the epididymis, with approximately 4 orders of magnitude less sperm recovered from the epididymis of *Tra2b-cPEko* mice versus wild type mice (Fig. 1J). These observations suggest that some isolated foci within *Tra2b-cPEko* testes remain able to complete spermatogenesis, although we cannot exclude that some germ cells escape *Vasa-Cre* mediated recombination.

The testes from *Tra2b-PEhet* mice showed no significant differences in testes weight or sperm count with wild type mice (Fig. 1E,J). Hence a single *Tra2b* allele with PE splicing control is sufficient to support successful spermatogenesis, so the wild type and *Tra2b-PEhet* genotypes are functionally equivalent in terms of germ cell development.

## Tra2β protein expression levels increase during the developmental window when germ cells die in the *Tra2b-cPEko* mice

Immunostaining of wild type testis sections showed that Tra2β expression is dynamically regulated during spermatogenesis. γH2AX

is a marker of DNA double strand breaks and is widely distributed during leptotene and zygotene, but by early pachytene become enriched on the un-synapsed X and Y chromosomes to form a focal concentration called the "sex body" (Mahadevaiah et al, 2001; Turner, 2007). Immunofluorescent analysis of wild type testis sections showed that leptotene cells with general nuclear staining of γH2AX expressed lower levels of Tra2β protein compared to pachytene cells that localised γH2AX within a sex body (Fig. 2A–C). Immunohistochemical analysis of wild type mouse testes confirmed high Tra2β protein levels during pachytene, the stage during which germ cells eventually die when reaching epithelial stage IV in *Tra2b-cPEko* mice (Fig. EV2A). In wild type mice, high levels of Tra2β expression are maintained up to diplotene stage, with weaker staining within the mitotically active spermatogonia and post-meiotic haploid round spermatids (Fig. EV2A). Less Tra2β protein expression was detected within the leptotene and zygotene meiotic sub-stages of spermatocyte development (Fig. EV2A). Further histological analyses of *Tra2b-cPEko* adult mouse testes confirmed that, besides becoming apoptotic, pachytene cells expressing high levels of Tra2β protein were also detaching into the lumen of their seminiferous tubules (Fig. EV2A).

## *Tra2b-cPEko* germ cells develop cellular and molecular defects

The above data indicate that Tra2β is significantly upregulated in pachytene cells compared to leptotene and zygotene cells. Germ cells with abnormalities can arrest during the pachytene stage of meiotic prophase, after which apoptosis is induced in aberrant cells that fail a quality check (called the stage IV checkpoint, Fig. 1B) (de Rooij and de Boer, 2003). To investigate in more detail the biological effects on meiotic prophase of deleting the *Tra2b* PE we monitored development of neonatal testes undergoing the first synchronous wave of spermatogenesis (Bellve et al, 1977). Histological analysis showed postnatal day 12 (P12) wild type and *Tra2b-cPEko* testes had similar cellular compositions (containing cells up to zygotene), although within the *Tra2b-cPEko* testes some zygotene cells already showed evidence of cell detachment into the lumen of seminiferous tubules (Fig. EV2B). Further histological analysis showed more extreme germ cell disruption of testes at postnatal day 14 (P14): while more pachytene cells had

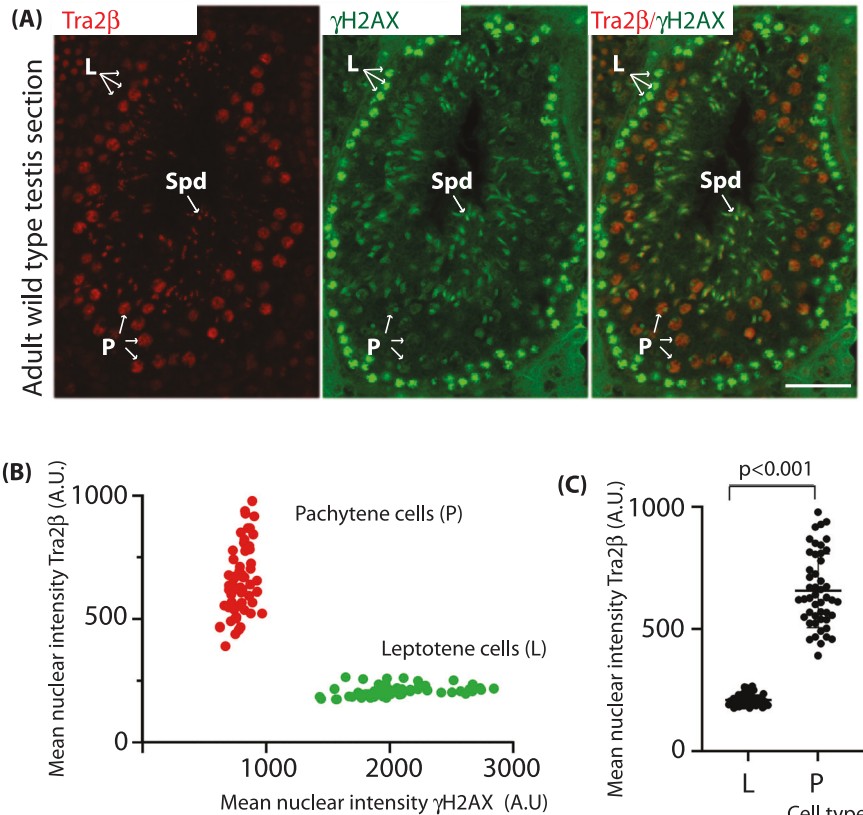

**Figure 2. Tra2β protein expression levels increase during the developmental window when germ cells die in the *Tra2b-cPEko* mice.**

(A) Indirect immunofluorescent image of mouse testis seminiferous tubule at stage 9–10 stained for Tra2β and γH2AX. Representative leptotene cells (labelled L), pachytene cells (labelled P) and elongating spermatids (labelled Spd) are arrowed. Scale bar = 50 μm ($n = 1$ wild type testis). (B) Quantitation of individual nuclear expression levels of Tra2β and γH2AX from the image shown in (A). Nuclear intensities from cells identified as pachytene are labelled red, and from cells identified as leptotene are labelled green. The individual pachytene and leptotene nuclei were identified visually after quantification ($n = 63$ leptotene nuclei, and $n = 61$ pachytene nuclei). (C) Comparison of Tra2β protein immunofluorescence levels in the individual pachytene and leptotene nuclei ($n = 63$ leptotene nuclei, and $n = 61$ pachytene nuclei). Plots show mean and standard deviation, and statistical significance calculated using a t test (Graphpad prism). Source data are available online for this figure.

appeared in the wild type testis, within the *Tra2b-cPEko* mice large numbers of germ cells had sloughed off into the lumen of seminiferous tubules (indicated by red arrows in Fig. EV2C).

Further analysis confirmed the presence of cellular defects within the *Tra2b-cPEko* testis. Immunofluorescent staining for γH2AX on P12 testis sections showed that in wild type testes ~75% of late zygotene cells contained distinct focal concentrations of γH2AX staining typical of a mature sex body, with the remaining ~25% having a less mature elongated "tadpole-shaped" sex body structure (Mahadevaiah et al, 2001) (Fig. 3A). In contrast, while *Tra2b-cPEko* P12 testes still contained cells with general nuclear distributions of γH2AX typical of leptotene and zygotene, ~90% of those cells that contained a sex body had less mature elongated "tadpole-shaped" sex bodies (Fig. 3A).

### *Tra2b-cPEko* P12 testes exhibit increased *Tra2b* gene expression and decreased expression of key meiotic genes

The above data showed that the *Tra2b-cPEko* germ cells manifest developmental abnormalities at the zygotene/pachytene boundary, followed by cell death and sloughing off during pachytene. The first

wave of mouse spermatogenesis is synchronised. We thus used RNAseq to monitor patterns of gene expression in whole *Tra2b-cPEko* and wild type testes taken at P12, in which the most advanced germ cells would be near the zygotene/pachytene boundary. Bioinformatics analysis of this whole P12 testis RNAseq data using DESeq2 (Love et al, 2014) identified 503 genes that were differentially expressed (with an adjusted *p* value of less than 0.05) between *Tra2b-cPEko* versus wild type testes (Fig. 3B and Fig. 3 Source data). These genes comprised 300 upregulated and 203 downregulated genes (red and blue dots respectively in Fig. 3B). These relatively similar numbers of upregulated and downregulated genes are consistent with the histological data (Fig. EV2B) that showed no changes in overall cell type composition between *Tra2b-cPEko* versus wild type testes at P12.

Notably, this RNA seq also indicated that whole P12 *Tra2b-cPEko* testes had upregulated *Tra2b* mRNA expression (Log2-fold expression change of 0.269, equivalent to ~20% increased expression, padjust = 0.00021, Fig. 3B). Quantitative indirect immunofluorescence showed nuclei within the P12 *Tra2b-cPEko* testis also had a significantly (Mann–Whitney U Test, $P < 0.0001$) higher intensity of Tra2β protein staining (median: 697.1, IQR: 560.6–845.4) compared to wild type (median: 547.27, IQR:

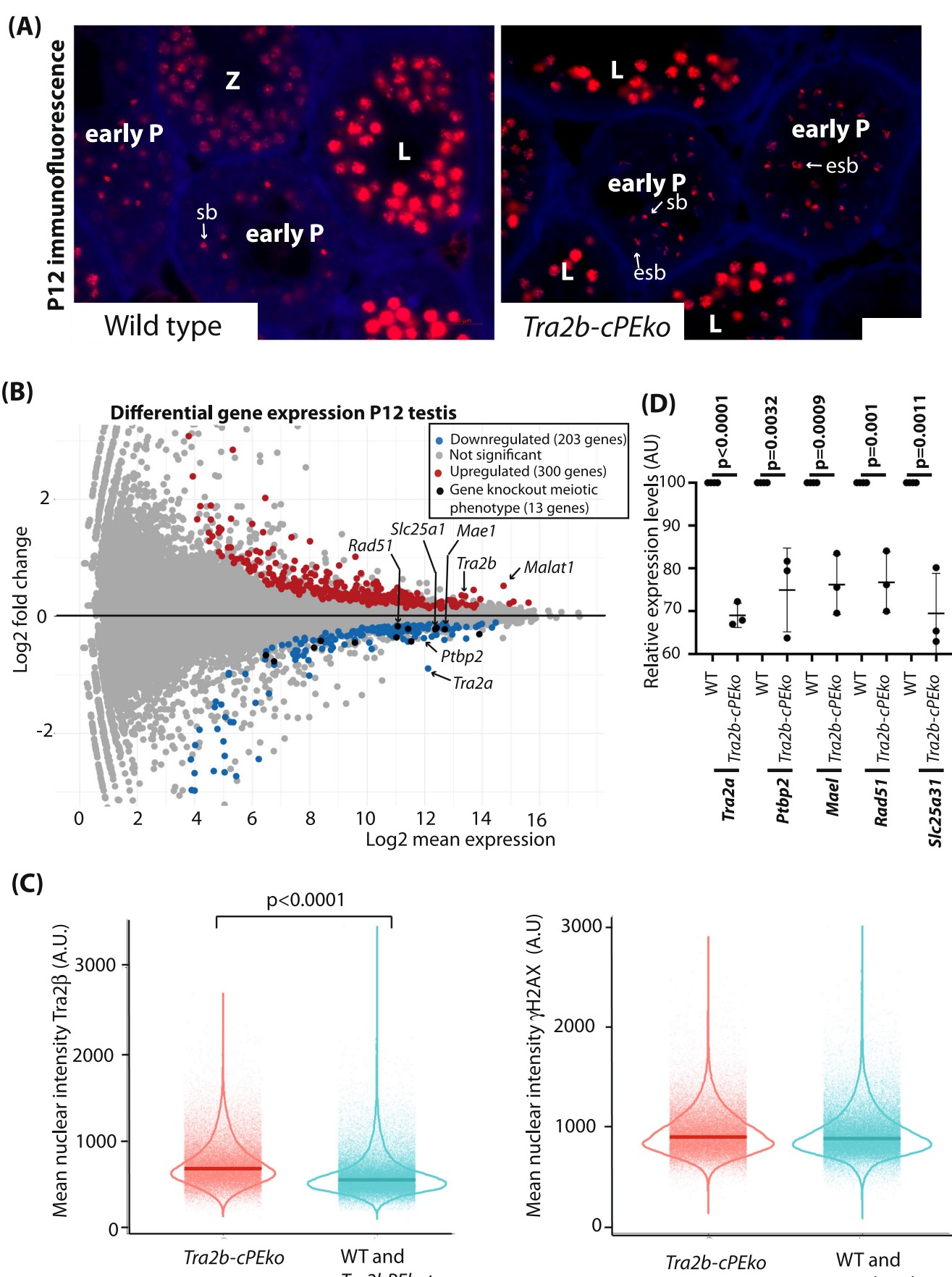

◀ **Figure 3. *Tra2b-cPEko* germ cells develop cellular and molecular defects.**

(A) Immunofluorescent detection of γH2AX on sections of P12 wild type and *Tra2b-cPEko* mouse testes (*n* = 4 of each genotype were analysed). Seminiferous tubules are labelled according to their cell content (L/Z: seminiferous tubules containing leptotene/zygotene cells; P: seminiferous tubules containing pachytene cells). Abbreviations: sb, sex body; esb, elongated sex body. Scale bar = 20 μm. (B) MAplot showing gene expression levels in P12 testis transcriptomes and how they change between testes from wild type and *Tra2b-cPEko* mice (using RNAseq data from *n* = 4 testes from each genotype, analysed using DESeq2). Genes with an adjusted *p* value of less than 0.05 (calculated by using the DESEQ2 default settings, that use a Wald test) are shown as red dots (if upregulated) or blue dots (if downregulated). Significantly downregulated genes that have been identified in mouse genetic analyses with an essential role in meiosis are shown as black dots (with further details in Fig. S2). Expression of all other genes are shown as grey dots. (C) Violin plot of Tra2β and γH2AX protein expression quantitation using indirect immunofluorescence of P12 testis sections (*n* = 331,667 identified and measured nuclei). Sections from 3 *Tra2b-cPEko* (shown in red) versus 1 wild type and 2 *Tra2b-PEhet* mice (shown in blue, chosen since these genotypes are both fertile) were analysed. Statistical significance was assessed using a Mann–Whitney U Test, and the median values are shown as a horizontal line. (D) Confirmation of predicted expression changes of a panel of genes between P12 wild type and *Tra2b-cPEko* testes using RT-qPCR (*n* = 4 per genotype). Pairwise tests of statistical significance were done using t tests, and the median line is shown in the scatter plot. Source data are available online for this figure.

458.83–667.66) (Fig. 3C). In contrast almost equal levels of γH2AX were detected within these same P12 *Tra2b-cPEko* (median: 889.6, IQR: 786.5–1048.7) compared to wild type and *Tra2b-PEhet* testis sections (median: 883.26, IQR: 779.32–1031.55) (Fig. 3C). Hence Tra2β protein is also increased by ~27% within the P12 *Tra2b-cPEko* testis that is unable to complete meiosis, compared to wild type and *Tra2b-PEhet* testes that are fertile and able to progress through meiosis. These data confirm that the *Tra2b* PE normally operates to reduce *Tra2b* mRNA expression in wild type germ cells. Although the NMD pathway does change activity during meiosis (*Upf3b* is on the X chromosome so becomes inactivated in the sex body), the NMD pathway would still be functional within most cells within a P12 testis, and so deplete PE-containing *Tra2b* mRNAs. However, even after meiotic sex body formation, PE splicing inclusion would still cause a frameshift and prevent productive expression of the *Tra2b* mRNA, even if the transcript was not also degraded by NMD.

Several of the downregulated genes within the P12 *Tra2b-cPEko* testes have previously been shown to be essential for male fertility, and specifically meiosis, in mouse knockout studies (shown as black dots in Fig. 3B and in more detail in Appendix Fig. S2) (Blake et al, 2021). These downregulated genes include genes important in chromosome biology such as *Maelstrom*, *Rad51* and *Mei1* (Soper et al, 2008; Dai et al, 2017; Libby et al, 2003), and global regulators of gene expression such as *Nxf2* (Pan et al, 2009). We further confirmed several of these genes are downregulated using RT-qPCR, including for *Maelstrom* and *Rad51* (Fig. 3D). Further Gene Set Enrichment Analysis (GSEA) indicated downregulation of genes involved in spermatogenesis in *Tra2b-cPEko* testes (Appendix Fig. S3A). Thus the reduced expression of some genes required for spermatogenesis may contribute to the failure of *Tra2b-cPEko* cells to proceed through pachytene.

### *Tra2b-cPEko* testes aberrantly splice Tra2β target exons

Tra2β is a potent splicing activator (Best et al, 2014a). Hence if Tra2β expression levels increase to levels causing cellular toxicity during meiosis, we would expect to detect associated aberrant splicing patterns in the *Tra2b-cPEko* testes. Consistent with this, further bioinformatic analysis using Leafcutter (Li et al, 2018) detected a panel of 157 genes with significant splicing differences, defined as having a padjust equal to or less than 0.05 and a ΔPSI (Percentage Splicing Inclusion) of greater than 0.1 between the wild type and *Tra2b-cPEko* P12 testes (Fig. 4A). Of these predicted splicing changes, 23 could be easily visualised on a genome browser

after alignment of RNAseq reads. We experimentally validated a sample of 4 activated exons within the *Ggnbp2*, *Rpgr*, *Map7d2*, and *Tra2a* genes by RT-PCR analyses on the 4 biological replicate wild type and *Tra2b-cPEko* P12 mouse testis RNA samples used for RNAseq (Fig. 4B). We could also detect the same splice changes in *Map7d2*, *Rpgr* and *Tra2a* between wild type and *Tra2b-cPEko* P14 mouse testes (Appendix Fig. S4A–E). These data confirm there are aberrant splicing patterns within the *Tra2b-cPEko* testis, and so confirm that some alternative exons are acutely sensitive to *Tra2b* PE deletion. GO analysis indicated enrichment of the 157 genes with predicted splicing changes in processes connected with spermatogenesis and meiosis (Appendix Fig. S3B).

To identify which of the splice events that change in the *Tra2b-cPEko* mouse testis are also targets for direct Tra2β protein binding we carried out iCLIP analysis (Konig et al, 2011) using wild type adult mouse testes (Fig. EV3A,B). Almost 27% of unique iCLIP tags mapped to exons (internal exons plus 5′ and 3′ untranslated regions), even though exons only make up 1% of the genome (Fig. EV3C). This iCLIP data also revealed that 18 of the 23 (78%) higher amplitude splicing changes had associated Tra2β iCLIP tags, indicating they are direct targets for Tra2β binding within adult mouse testis. These included the *Tra2a* gene PE, that contained multiple iCLIP tags and was also previously known to bind to and be activated by Tra2β (Grellscheid et al, 2011a; Best et al, 2014b). Strikingly, the most significant splicing difference detected between wild type and *Tra2b-cPEko* testes was also increased inclusion of this *Tra2a* PE (Fig. 4A–C). PE inclusion should de-stabilise the *Tra2a* mRNA, and exactly consistent with this molecular fingerprint, downregulation of *Tra2a* was detected as the most significant change in overall gene expression within the P12 transcriptome (Fig. 3B), equivalent to a 47% reduction (padjust = 2.41E-23). We further confirmed this downregulation of *Tra2a* expression using RT-qPCR (Fig. 3D). We also detected increased inclusion of the *Nasp-T* exon that had also previously been identified as a molecular target for Tra2β binding, and is a protein-coding cassette exon (Fig. 4A) (Grellscheid et al, 2011a; Best et al, 2014b). Further novel Tra2β target exons that changed splicing in the *Tra2b-cPEko* P12 mouse testis included exon 6 of the *Ggnbp2* gene that had a high concentration of iCLIP tags, and is also protein-coding (Figs. 4A and EV4A,B). *Ggnbp2* is required for spermatogenesis and encodes a protein involved in RNA turnover required for meiotic chromosome break repair during meiosis (Guo et al, 2018, 2024; Mauxion et al, 2023). Increased inclusion of a direct Tra2β-target exon was also detected within the *Rpgr* gene (protein-coding exon 14, Figs. 4A,B and EV4C), expression levels of

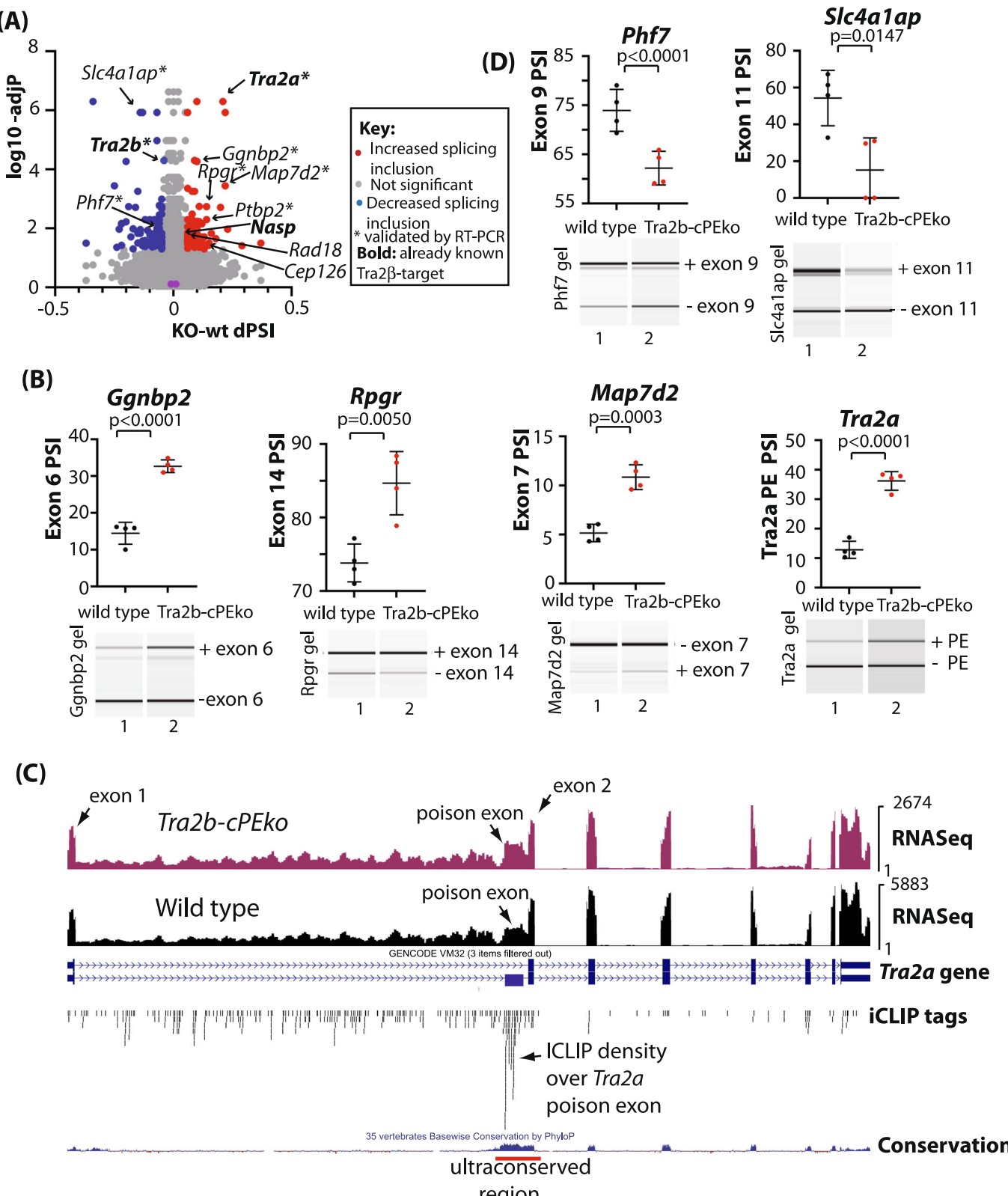

**Figure 4.  *Tra2b-cPEko* testes display aberrant splicing patterns of Tra2β target exons.**

(A) Volcano plot showing results of differential splice isoforms detected using Leafcutter (Li et al, 2018). Splice isoform changes with a p.adjust equal or less than 0.05 and a ΔPSI of greater than 0.1 are shown in either red (upregulated) or blue (downregulated), with remaining splice isoforms shown in grey (calculated by default settings in Leafcutter, that uses a Likelihood ratio test). Note that Leafcutter outputs a number of datapoints for each gene corresponding to the ΔPSI of each mRNA isoform detected. Some individual genes with high amplitude splice isoform switches in response to PE deletion are labelled. (B) Experimental confirmation of splice isoforms activated within P12 *Tra2b-cPEko* testes using RT-PCR analysis. Lower panels: Representative capillary gel electrophoretograms. Upper panels: Data from 4 biological replicates of each genotype also used for RNAseq. The mean PSI is shown as a bar with SD as error bar, and p values were calculated using a t test. Note that *Map7d2* uses an additional internal primer, so the lower band represents exon skipping. Independent confirmation of some of these splice changes within independent P14 wild type and *Tra2b-cPEko* testes are shown in Appendix Fig. S4. (C) *Tra2b* PE deletion changes splicing patterns of the *Tra2a* gene. UCSC genome browser screenshot showing P12 testis RNAseq reads and adult testis Tra2β iCLIP tags aligned to the *Tra2a* gene on the mouse genome (mm39). (D) Experimental confirmation of splice isoforms repressed within P12 *Tra2b-cPEko* testes using RT-PCR. Lower panels: Representative capillary gel electrophoretograms. Upper panels: Data from 4 biological replicates of each genotype. The mean PSI is shown as a bar with SD as error bar, and p values were calculated using a t test. Independent confirmation of some of these splice changes within independent P14 wild type and *Tra2b-cPEko* testes are shown in Appendix Fig. S4. Source data are available online for this figure.

which can impact spermatogenesis (Brunner et al, 2008); *Rad18* (protein-coding exon 10, Fig. 4A) which is involved in sex body formation during meiotic prophase (Inagaki et al, 2011); and an alternative exon within intron 5 of the *Cep126* gene (Fig. 4A) that encodes a centrosomal component involved in mitosis (Bonavita et al, 2014). The splicing changes detected within the *Nasp*, *Rpgr* and *Map7d2* genes were within exons encoding disordered protein domains.

Not all of the splicing changes in the P12 *Tra2b-cPEko* testes involved increased inclusion of cassette exons. Our bioinformatics search also detected decreased inclusion of the *Tra2b* PE (corresponding to genetic deletion of this exon in our *Tra2b-cPEko* mouse model) (Fig. 4A; Appendix Fig. S5). We also detected decreased inclusion of exon 9 of the *Phf7* gene (Fig. 4A,D) and exon 11 (encoding a disordered protein domain) of the *Slc4a1ap* gene (Fig. 4A,D). This indicates that both these protein-coding alternative exons are skipped in response to increased levels of Tra2β protein. Interestingly iCLIP also identified strong Tra2β binding to the non-coding RNA *Malat1*, gene expression of which was detected as significantly upregulated in the *Tra2b-cPEko* testis (Fig. 3B; Appendix Fig. S6). *Malat1* has previously been found to bind to Tra2β and other SR proteins, but the function of this binding has been unclear. No changes in *Malat1* splicing were detected in the *Tra2b-cPEko* testes, but the increased *Malat1* gene expression in the *Tra2b-cPEko* mice is consistent with higher levels of Tra2β protein binding increasing *Malat1* stability.

### *Tra2b-cPEko* testes show increased cryptic splicing in the *Ptbp2* gene

In addition to previously annotated alternative spliced exons, cryptic splicing events within the *Ptbp2* gene were detected in the *Tra2b-cPEko* testes. *Ptbp2* encodes the splicing regulator protein PTBP2 that is essential for mouse germ cell development, with key roles during and after meiosis (Hannigan et al, 2017). A high density of Tra2β iCLIP tags mapped to *Ptbp2* introns 8 and 9, indicating these introns are directly bound by Tra2β protein, a feature that might cause them to respond to concentration increases of Tra2β protein. Consistent with this, we also detected higher levels of a previously undescribed splice junction joining *Ptbp2* exon 8 to a novel cryptic 3′ splice site within intron 8 (Fig. 5A). Isoform-specific RT-PCR confirmed increased use of this cryptic splice junction within *Tra2b-cPEko* compared to wild type testes in P12 testes (Fig. 5B), and we also detected increased use of

this cryptic splice site within P14 *Tra2b-cPEko* compared to wild type P14 testes (Appendix Fig. S4D). No signal was obtained when we ran this same analysis without reverse transcriptase, confirming it was detecting RNA rather than a signal from any contaminating DNA (Appendix Fig. S4F).

Further analysis indicated these changes in *Ptbp2* splicing are likely to be functionally important for gene expression. Global DESeq2 analysis (Fig. 3B) showed that *Ptbp2* gene expression was significantly decreased in P12 *Tra2b-cPEko* compared to wild type testes (equivalent to a 23% reduction, padjust = 0.000332). We further confirmed downregulation of *Ptbp2* expression within P12 *Tra2b-cPEko* using RT-qPCR (Fig. 3C). The increased selection of novel Tra2β-dependent *Ptbp2* splice isoforms within *Tra2b-cPEko* testis (utilising the cryptic splice) may thus contribute to reducing *Ptbp2* gene expression by targeting *Ptbp2* mRNA for NMD.

### Activated splice sites in the *Tra2b-cPEko* testes are hypersensitive to Tra2β protein concentrations

The above data showed that the splicing targets which had the most activated splicing within the *Tra2b-cPEko* testes were also direct targets for Tra2β protein binding, and often contained high concentrations of Tra2β binding sites. This predicted a model where high levels of Tra2β binding sites would render particular splice sites hypersensitive to increased expression of Tra2β protein within the *Tra2b-cPEko* testes (Fig. 5C). We tested this model further using a minigene approach. Although minigene experiments sometimes do not perfectly analyse splicing in the context of normal upstream and downstream splice sites, previous minigene analyses showed that both the *Nasp-T* exon and *Tra2a* poison exon were highly sensitive to Tra2β protein concentrations in vitro (Grellscheid et al, 2011a). We further constructed a minigene that included a large segment of *Ptbp2* intron 8 including the novel intronic cryptic splice site flanked by β-globin exons (Fig. 5D), and analysed splicing patterns from this minigene in HEK293 cells. When co-transfected with GFP, a splice product corresponding to use of the cryptic splice site was not detected, showing this cryptic splice site is not selected at ambient cellular concentrations of Tra2β (Fig. 5E–G, lane 1). However, RT-PCR analysis clearly detected the cryptic splice product after co-transfection of the minigene with an expression vector encoding a Tra2β-GFP fusion protein (Fig. 5E–G, lane 2). Indicating direct RNA-protein contact is important for this aberrant regulation, no cryptic splice site activation was observed after co-transfection of the minigene with

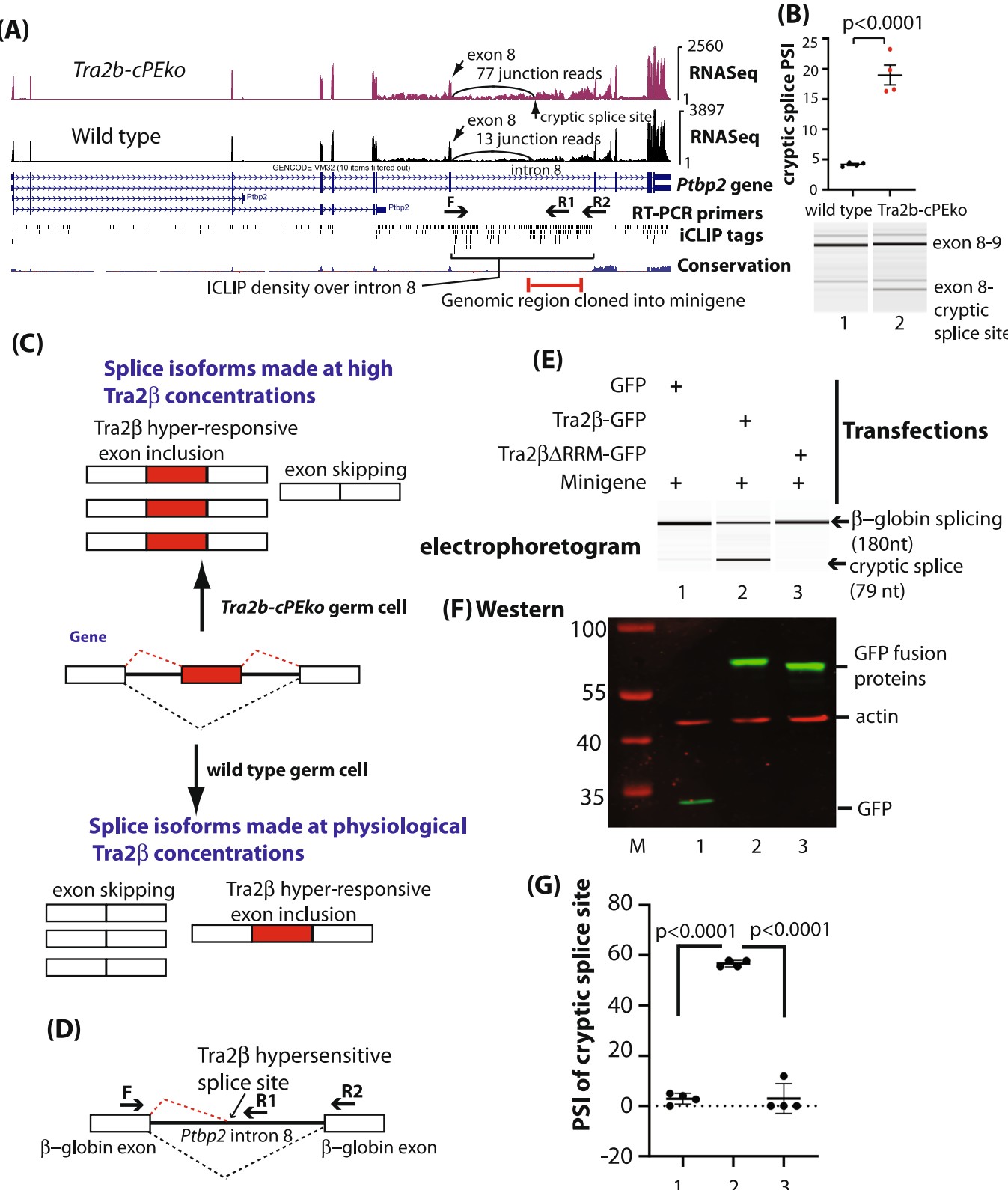

**Figure 5.   Activated splice sites in the *Tra2b-cPEko* testes are hypersensitive to Tra2β protein concentrations.**

(A) UCSC genome browser screenshot of the mouse genome (mm39), showing P12 testis RNAseq reads and Tra2β iCLIP tags aligned to the *Ptbp2* gene, including the position of the cryptic splice site within intron 8. (B) Detection of cryptic splice selection using RT-PCR (lower panel: representative capillary gel electrophoretogram; upper panel: data from 4 biological replicate mice from each genotype showing mean percentage splicing inclusion $+/-$ SEM). (C) Model predicting behaviour of Tra2β hyper-responsive exons in *Tra2b-cPEko* germ cells. (D) Design of *Ptbp2* intron 8 minigene to test behaviour of a candidate Tra2β hyper-responsive splice site. The relative position of primers (F, R1 and R2) used for the RT-PCR analysis are shown. (E) Representative capillary electrophoretogram showing detection of cryptic splice isoforms in RNA isolated from HEK293 cells after transfection with the *Ptbp2* intron 8 minigene and expression constructs encoding either GFP or Tra2β-GFP-fusion proteins (from $n = 4$ biological replicates). (F) Corresponding Western blot analysis probed for GFP of protein extracted from HEK293 cells after transfection of expression constructs and minigenes (from $n = 4$ biological replicates). (G) Graph showing mean PSI levels of the Tra2β hyper-responsive splice site within each group of transfected HEK293 cells. In each case, data were generated from 4 biological replicates and error bars represent SD. *P* values were calculated using t tests. Source data are available online for this figure.

an expression vector encoding a version of Tra2β lacking its RRM (Tra2βΔRRM-GFP, Fig. 5E–G, lane 3). Parallel Western blots showed that the Tra2βΔRRM-GFP protein was expressed in HEK293 cells at similar levels to Tra2β-GFP (Fig. 5F). These data support a molecular model in which cryptic splice site activation in *Ptbp2* responds to increased expression of Tra2β protein in the *Tra2b-cPEko* testis, and for which Tra2β protein-RNA molecular interactions are important.

## The Tra2b PE is not required by mitotically proliferating male germ cells

The above data indicated that the *Tra2b* PE is essential in spermatogenesis during meiotic prophase. However, recombination mediated by *Vasa-Cre* within the mouse germline is complete by birth (Gallardo et al, 2007). This temporal difference implied that the spermatogonial cell population (that includes the mitotically active stem cells, some of which differentiate into spermatocytes, Fig. 1B) must be able to survive without the *Tra2b* PE and thus the *Tra2b* homeostatic feedback loop. However, an alternative explanation for the survival of spermatogonia in the *Tra2b-cPEko* testis could be that meiotic prophase is also the first step of spermatogenesis that requires the entire *Tra2b* gene. To test between these alternative possibilities we utilised the same *Vasa-Cre* transgene to inactivate an existing conditional *Tra2b* allele (Mende et al, 2010), in which *Tra2b* exon 4 is flanked by *LoxP* sites (*Tra2b^fl*, Fig. 6A) (Mende et al, 2010; Gallardo et al, 2007). Experimental mice were generated with the following genotypes: *Tra2b^fl/+* (hereafter referred to as wild type); *Tra2b^fl/-* and *Tra2b^fl/+;Vasa-Cre* (both hereafter referred to as *Tra2b-het*); and *Tra2b^fl/-;Vasa-Cre* (hereafter referred to as *Tra2b-cko*). Details of the crosses used to generate our experimental mice are given in Fig. EV5A–C.

Analysis of these mice detected a very early requirement during germ cell development for the *Tra2b* gene, within the spermatogonial cells. Adult *Tra2b-cko* mice had considerably smaller testes compared to those from their wild type littermates (Fig. 6B,C) and were azoospermic (Fig. 6D). Histology showed a complete absence of male germ cells in *Tra2b-cko* adult testes (Fig. 6E). Despite this adult "Sertoli cell only" phenotype, immunohistochemical detection of the germ cell-specific RBMY protein (Saunders et al, 2003) within neonatal mouse testes showed that *Tra2b-cko* mice were born with similar numbers of male germ cells relative to their wild type littermates (Fig. 6F). However, whereas wild type germ cells increased in number between days P0–P3 (postnatal days 1–4), the numbers of RBMY-positive germ cells in *Tra2b-cko* mouse testes

progressively reduced over this same developmental window. These data show that the small testis phenotype in adult *Tra2b-cko* mice is caused by reduced proliferation and demise of the spermatogonia. The observation that the *Tra2b* gene is required for mitotic proliferation of spermatogonia is consistent with analyses in cultured human cell lines where CRISPR inactivation of the *TRA2B* gene reduces mitotic cell proliferation (Thomas et al, 2020).

Interestingly, these experiments further confirmed that germ cell development is profoundly sensitive to expression levels of *Tra2b*. Adult testes from *Tra2b-het* mice that lack one copy of *Tra2b* were significantly smaller than wild type testes and had reduced sperm counts (Fig. 6B–D). This exquisite dose-sensitivity is consistent with *Tra2b* gene expression levels directly impacting reproductive success within wild populations, where females might likely have more than one breeding partner making sperm production levels important (Parker, 2020).

Previous work has shown that Vasa-Cre mediated recombination initiates at E15 and is largely complete by birth (Gallardo et al, 2007). To confirm this, we monitored the efficiency of Vasa-Cre mediated recombination within our mice. Nuclear Tra2β protein expression was detected within both spermatogonia and Sertoli cells of wild type postnatal day 1 (P1) testes (Fig. 6G, left panel). In contrast, Tra2β protein was absent from spermatogonia at P1 in *Tra2b-cko* mice, but easily detectable within adjacent Sertoli cells (Fig. 6G, right panel). These data are consistent with the *Vasa-Cre* transgene having efficiently inactivated *Tra2b* gene expression within the germline by P1. Female *Tra2b-cko* mice were also infertile, although these were not further analysed in this study (Fig. EV5D).

## Discussion

In summary, the above experiments show that the *Tra2b* PE is essential for male germ cells during meiotic prophase and ultimately for sperm production and male fertility. *Tra2b-cPEko* germ cells progress normally through male germ cell development, until they reach the pachytene stage where they fail the Stage IV quality control checkpoint and undergo apoptosis (de Rooij and de Boer, 2003). Our data suggest a model where the *Tra2b* PE splicing dampens Tra2β expression to reduce selection of splice sites with high associated densities of Tra2β binding sites, causing hypersensitivity to increased expression of Tra2β protein within the *Tra2b-cPEko* testes. This is consistent with previous minigene data showing that Tra2β binding site density correlates with Tra2β sensitivity (Grellscheid et al, 2011b, 2011a; Best et al, 2014b). Germ

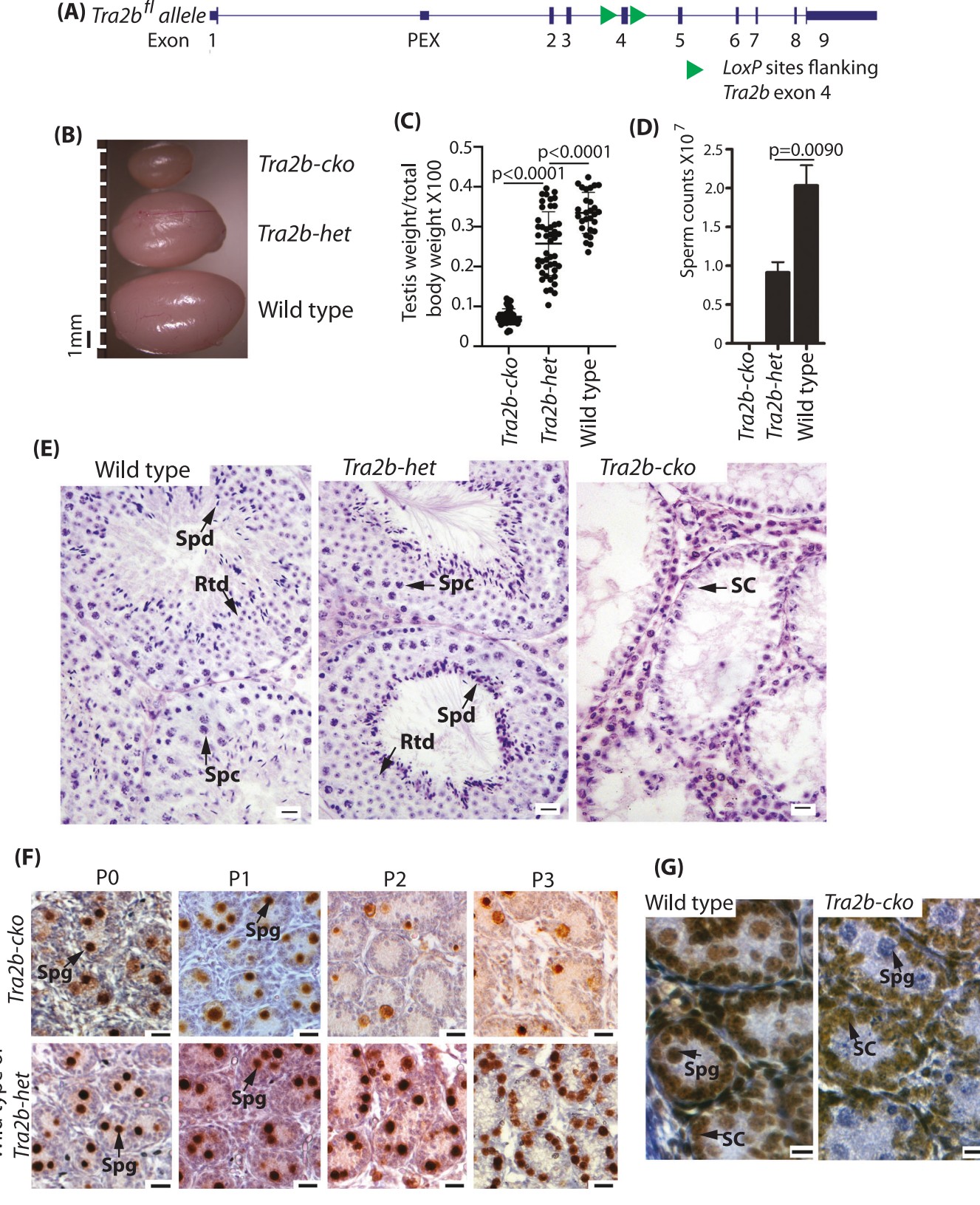

**Figure 6.** *Tra2b* gene function is required for mitotic proliferation within the germline.

(A) Schematic of the *Tra2b* exon 4 conditional allele showing the position of the *LoxP* sites used to create the *Tra2b-cko* mouse line. (B) Morphologies of adult testes from different genotype mice (scale shown beside testes is in millimetres). (C) Adult testis:body weight ratios of different genotype mice. Individual values are shown, with the mean and SD indicated. *P* values were calculated using a t test (n = 46 *Tra2b-cko* testes, n = 46 *Tra2b-het* testes, n = 28 wild type testes, [age range 4–33 weeks]). (D) Epididymal sperm counts from different genotype mice (n = 8 *Tra2b-cko*, n = 4 *Tra2b-het*, n = 3 wild type, age range 10–18 weeks). *P* values were calculated using a t test. The error bar represents the SEM. (E) Micrographs of hematoxylin-stained adult testis sections from different germline genotypes (from n = 3 per genotype). Scale bar = 20 μm. Abbreviations for cell types shown in Fig. 1B, with the addition of SC (Sertoli Cell). (F) Histological analysis of testis sections from *Tra2b-cko* (P0–P3), and wild type (P0–P2) or *Tra2b-het* (P3) neonatal mice. Testes were harvested between the day of birth (P0) and postnatal day 3 (P3), stained for RBMY protein and counterstained with hematoxylin (Abbreviations for cell types are shown in Fig. 1B). Sample numbers: P0 (n = 2 *Tra2b-cko*, n = 2 wild type); P1 (n = 2 *Tra2b-cko*, n = 2 wild type); P2 (n = 2 *Tra2b-cko*, n = 3 wild type); P3 (n = 2 *Tra2b-cko*, n = 1 *Tra2b-het*, n = 1 wild type). Scale bar = 20 μm. (G) Immunostained P1 testis sections stained for Tra2β protein, and counterstained with hematoxylin (from n = 2 *Tra2b-cko* and n = 2 wild type). Abbreviations for cell types shown in Fig. 1B, with the addition of SC (Sertoli Cell). Scale bar = 20 μm. Source data are available online for this figure.

cell development requires carefully orchestrated patterns of gene expression (Soumillon et al, 2013), and the splicing abnormalities induced by extra Tra2β expression could delay proper developmental patterns needed for meiotic progression. A number of the splice changes in *Tra2b-cPEko* testes are within genes important for germ cell development, but whether defects in one gene in particular causes the phenotype or a combination requires further analysis.

Creation of separate *Tra2b-cPEko* and *Tra2b-cko* mouse models using the same *Vasa-Cre* (active from E15, so causing deletion within embryonic germ cells from which all later stages of germ cell development arise) cause different infertility phenotypes. This indicates both expression and expression control (via PE splicing) of Tra2β expression are required for spermatogenesis, but during different temporal windows. Tra2β expression is needed for survival of mitotically proliferating spermatogonia, with *Tra2b* inactivation resulting in spermatogonia cell death and a "Sertoli Cell Only" phenotype. This is analogous to the situation in cultured cells, where the *Tra2b* gene is also required for mitotic proliferation. Despite their requirement for Tra2β protein, *Tra2b-cPEko* spermatogonia are able to proliferate via mitosis without the *Tra2b* PE, but develop into *Tra2b-cPEko* spermatocytes that then die, resulting in a meiotic arrest phenotype. Since the human *TRA2B* PE is also not essential for mitotic proliferation in cultured cells (Thomas et al, 2020; Leclair et al, 2020) and deletion of the human TRA2B PE promotes mitotic T cell proliferation (Karginov et al, 2024), this essential role in meiotic prophase was unexpected. However, unlike mitosis, meiotic prophase I is highly transcriptionally active (Aldalaqan et al, 2022). Pachytene cells might be particularly vulnerable to *Tra2b* PE deletion because of their increasing endogenous levels of Tra2β expression, or might be more sensitive to proper patterns of splice isoforms regulated by normal concentrations of Tra2β.

Recent findings on the global occurrence of "unproductive splicing" (Fair et al, 2024) coupled with the role of the Tra2bPE in T cell receptor signalling (Karginov et al, 2024), together with the completely in vivo study reported here, are consistent with a broad and fundamental role for poison exons in fine-tuning precise, tightly-regulated, developmental processes. In particular, the association of the *Tra2b* PE with male fertility provides a compelling link with its genomic ultra-conservation. Spermatogenesis produces ~45 million sperm daily/testis and is strongly exposed to selective pressure (Griswold, 2016; Murat et al, 2023), meaning that genetic changes that reduced PE function would be strongly selected against. Our data also highlight the power of in vivo loss of

function analysis to fully understand ultra-conserved genetic regions and the evolutionary pressures that constrain these elements, and show that the ultra-conserved SR protein gene PEs have fundamental developmental functions exceeding those of ultra-conserved enhancer sequences analysed to date (Snetkova et al, 2022). A further implication of our data is that the *Tra2b* PE and PEs in other SR protein genes may contribute more widely to extensive, exquisitely sensitive, feedback mechanisms controlling gene expression levels at other stages of development, and form part of a widespread and fundamental regulatory mechanism conserved across kingdoms (Lareau et al, 2007; Lareau and Brenner, 2015). The conditional *Tra2b* PE allele we describe here is thus a key reagent that can be used more widely to test the involvement of the *Tra2b* PE in other developmental stages and processes.

## Methods

### Generation of germ cell-specific *Tra2b* exon deletions and mouse phenotyping

The *Tra2b* gene was conditionally inactivated using an existing allele (Mende et al, 2010), with exon 4 flanked by *LoxP* sites. CRISPR was used to generate the C57BL/6J-Tra2b$^{em1H/H}$ strain that contains the *Tra2b* PE flanked by *LoxP* sites using the methods described by Codner et al (Codner et al, 2018). C57BL/6J-Tra2b$^{em1H/H}$ mice were made by pronuclear injection of sgRNAs (Table EV1: these gRNA sequences map to sequences outside of the ultra-conserved regions) and a long ssDNA donor sequence into a C57BL6/J background 1-cell stage embryo (Codner et al, 2018). Microinjection buffer (MIB; 10 mM Tris–HCl, 0.1 mM EDTA, 100 mM NaCl, pH 7.5) was prepared and filtered through a 2 nm filter and autoclaved. Cas9 mRNA, sgRNAs and ssODNs were diluted and mixed in MIB to the working concentrations of 100 ng/μl, 50 ng/μl each and 50 ng/μl, respectively. Injected embryos were re-implanted in CD1 pseudo-pregnant females. Host females were allowed to litter and rear F$_0$ progeny. Mice were maintained on a C57BL6/J background. To check for integration of the *LoxP* sites Genomic DNA was extracted from ear clip biopsies and amplified in a PCR reaction using the following conditions/primer sequences: Geno_Tra2b _F1 (5′-GTTCCAACAGTGTTCCAGTTTGT-3′ and 5′-TAGTTCAGTTGGAACAGGCGTT-3′ and Geno_Tra2b _R1 (wild type product size = 1752; targeted allele = 1756); LoxPF (5′-ATCCGGGGGTACCGCGTCGAG-3′) and LoxPR (5′-ACT

GATGGCGAGCTCAGACC-3′) (expected size = 1193); and a genomic PCR from the LoxP site to the external primers (LoxPF and Geno_Tra2b _R1, expected size = 1520; and Geno_Tra2b _F1 and LoxPR, expected size = 1429). All amplicons were Sanger sequenced to confirm insertion of the donor oligo sequence at the target site. Copy counting of the donor sequence was carried out by ddPCR at the F1 stage to confirm donor oligos were inserted once on target into the genome using the following Taqman assays to copy count the donor sequence compared against a VIC-labelled reference assay for *Dot1l* as per Codner et al (Codner et al, 2018): (1) Tra2b-FLOX-DONOR-MUT1 assay: Forward primer (5′-CGAT CGCATAACTTCGTATAGCATACAT-3′), reverse primer (5′- GGACAACTCATCTGCACAATGATG-3′), probe (ACACAGTAA TGTGGCATTAAATCCATTTCC); label (FAM-BHQ1). (2) Tra2b-FLOX-3′-MUT1 assay: forward primer (5′-GGCATGAATGAGTA TGAGTTGGAAC-3′), reverse primer (5′-AGTCCCATTTTCACC CTGxGTAAC-3′), probe (5′-AAGTTATCGCCGGCGGGTCTGA-3′), label (FAM-BHQ1). (3) Tra2b-CR-LOA assay: Forward primer (5′GCGGAAGTCGTCATTTGACAAG-3′), reverse primer (5′-TC CCCACTTCACACAAATTGCT-3′), probe (5′-TTGAAGCTCAGG AATAAGTGAAGCTGA-3′), label FAM-BHQ1. No evidence of random donor insertions were detected in the animals taken forward to establish the colony.

Conditional inactivation was performed using *Vasa-Cre* expressing mice (23) purchased from The Jackson Laboratory (JAX stock#006954) on an FVB background and backcrossed for 9 generations to C57BL/6J. The mouse alleles used in this study are shown in Table EV2. All mice were maintained on a C57BL/6J background. Experimental genotypes were generated via the crosses shown in Figs. EV1 and EV5.

Genotyping was performed by PCR of genomic DNA extracted from ear clips using the HotSHot method (Truett et al, 2000). Primer sequences and amplicon sizes can be found in Table EV3.

All mice were maintained on a C57BL/6J background under project licenses granted by the UK Government Home Office, following the guidelines and regulations dealing with the care and use of laboratory animals outlined by the Animals (Scientific Procedures) Act 1986. All mice were housed in Allentown Nexgen Individual Ventilated Cages (IVC), bedding on aspen chip grade 2 and nesting with 'sizzle nest' from Datesand. Diet was supplied via Envigo (Teklad 2019 extruded diet) and water was filtered to 1.0 micron. The temperature was controlled between 20 and 24 °C, and the humidity between 45 and 65%, with a 12/12 light cycle. Animals were health screened quarterly in line with FELASA recommendation. All work was approved by the Animal Ethics Committee of Newcastle University.

### Fertility analysis

For fertility analysis, engineered male mice (aged 7–11 weeks) were housed with wild type female mice (aged 8–10 weeks) and subsequent litter sizes were counted. Testes body/weight ratios were calculated and epididymal sperm counts made using a hemocytometer (as described in https://phenome.jax.org/projects/Handel1/protocol).

### Histology, immunohistochemistry and immunofluorescence

Dissected mouse testes were fixed in Bouin's and embedded in paraffin wax as previously described (Ehrmann et al, 2019). Histological analyses used 5 μm sections stained with hematoxylin or by periodic acid-Schiff (Ehrmann et al, 2019).

Immunohistochemical detection of Tra2β and RBMY proteins was carried out following an antigen retrieval step in 0.01 M pH 6.0 citrate buffer as previously described (Elliott et al, 1997) with the addition of signal amplification using TSA Biotin Kit (Perkin Elmer NEL700A001KT). Primary antibodies used were anti-TRA2B/ SFRS10 antibody (Abcam ab31353) and anti-RBMY (Saunders et al, 2003), and an HRP-conjugated anti-rabbit secondary antibody (Dako E0432). Standard immunofluorescent detection of anti-γH2AX also used an antigen retrieval step in 0.01 M pH 6.0 citrate buffer as previously described (Elliott et al, 1997), using primary anti-γH2AX (Sigma-Aldrich 05-636-I) and secondary anti-mouse Alexa Fluor® 594 (Thermofisher A10040).

### Quantitative indirect immunofluorescence

Antigen retrieval for quantitative immunofluorescence of Tra2β and γH2AX on Bouin's fixed mouse testis sections was carried out in Tris-EDTA buffer (10 mM Tris base; 1 mM EDTA; 0.05% Tween 20) overnight at 60 °C. These latter slides were also pre-stained with 0.25% Sudan black B for 90 min to quench autofluorescence. Fluorescent micrographs were captured using the same preset exposures to collect panelled images of entire 5 μm mouse testis sections from three independent P12 *Tra2b-cPEko* mice, and 1 wild type P12 mouse and 2 *Tra2b-cPEhet* mice (both fertile genotypes that had identical morphologies). Images were processed using a custom FIJI Macro (Schindelin et al, 2012), available at https:// github.com/NCL-ImageAnalysis/An-ultra-conserved-poison-exon- in-Tra2b-is-essential-for-male-fertility-and-meiotic-cell-division/ tree/main. Due to poor nuclear signal-to-noise, the channels for Tra2β and γH2AX were combined using the Enhance Contrast function (saturated = 0.1) followed by combining into a maximum intensity projection. A custom Ilastik model (Berg et al, 2019) was generated and trained on small cropped regions of the maximum intensity images generated to identify nuclear and non-nuclear pixels. The Ilastik model (available upon request due to file size limitations of GitHub) was applied to the full-scale images generated in the FIJI Macro. Resulting probability maps for nuclear identity were scaled by a factor of 0.5 to facilitate StarDist (Schmidt et al, 2018) use, with a probability threshold of 0.05 and a nms threshold of 0.25. Output binary representations of identified nuclei were scaled to the original size, and nuclei larger than 78 μm$^2$ (750 pixels) were excluded from further analysis. Intensity measurements for each nucleus were taken in each of the channels of the original images and exported in csv format for data processing and analysis. Imaging data was analysed using RStudio, the code for which is available at http://www.rstudio.com/, version used 2023.12.1.402) using code available within the github repository mentioned above. Summary analyses were generated and statistical analyses were performed using a non-parametric Mann–Whitney test.

### RNAseq and bioinformatics

Mouse numbers for RNAseq were based on encode recommendations for RNA sequencing (standards, guidelines and best practices for RNA-seq, v1.0, June 2011, the ENCODE consortium). A sample size of 4 wild type and 4 *Tra2b-cPEko* P12 testes from biologically independent mice were selected. Our experimental approach was to collect 12-day-old testes, so depending on the exact age there could

be some subtle cell type variation. No blinding was done as we needed to know sample IDs for bioinformatics processing. Total RNA was extracted from isolated testes using ReliaPrep RNA Tissue Miniprep System (Promega Z6111). Paired-end RNA sequencing was done for eight samples in total (four biological replicates of 12dpp wild-type and *Tra2b-cPEko* testes) using 100 bp reads. Libraries were prepared using TruSeq Stranded mRNA Library Prep Kit (Illumina) and analysed on a Novaseq machine (Illumina, using the mid output v2 150 cycles kit).

The bioinformatics processing and analysis performed in this work used the GRCm39 reference genome and its genomic annotation, which were downloaded from NCBI Assembly (https://www.ncbi.nlm.nih.gov/assembly/GCF_000001635.27/) as FASTA and GTF files, respectively.

After quality control of the FASTQ files using FastQC (http://www.bioinformatics.babraham.ac.uk/projects/fastqc/), each sample's reads were aligned using the HISAT2 suite of tools (Kim et al, 2019). Exons and splice sites were extracted from the GTF file using 'hisat2_extract_exons.py' and "hisat2_extract_exons.py", the outputs of which were used along with the reference FASTA file to produce an index with 'hisat2_build'. The resulting HISAT2 index was used with 'hisat2', producing alignments in SAM format, which were converted to BAM format using samtools (Li et al, 2009). The bamCoverage from the deepTools suite was used on the BAM files to produce bigWig files that enable visualisation of genome coverage with the UCSC genome browser (Nassar et al, 2023).

The quantification of transcriptomic reads was generated using Salmon (Patro et al, 2017) by feeding it each sample's FASTQ files. The produced read count tables were used as input to the R package DESeq2 (Love et al, 2014) to perform the differential gene expression analysis discussed in this work. These resulting data were visualised in plots generated by ggplot2 and ggpubr (Wickham, 2016).

The differential splicing analysis was performed using Leafcutter (Li et al, 2018). For the analysis of splice sites, Leafcutter first requires junction files extracted from the BAM files. In this work, the junction files were produced using regtools (Cotto et al, 2023) as recommended by the developers of Leafcutter. These junction files were then fed to 'leafcutter_cluster_regtools.py' to generate intron cluster counts for the different samples. These count tables were finally fed to 'leafcutter_ds.R' to perform the differential splicing analysis discussed in this work.

GSEA analysis was performed on RNA-sequencing data using GSEA_4.2.3. Analysis was run using the Hallmarks Gene Set with 1000 permutations (Subramanian et al, 2005). Gene Ontology analysis was run using PANTHER (Gene ontology database https://doi.org/10.5281/zenodo.10536401 Released 2024-01-17) on differentially spliced genes identified by RNA-sequencing following *Tra2b* poison exon deletion. Some phenotype data for this paper were retrieved from the Mouse Genome Database (MGD), Mouse Genome Informatics, The Jackson Laboratory, Bar Harbor, Maine. World Wide Web (URL: http://www.informatics.jax.org).

## Analysis of splice isoforms in mouse testis RNA

P12 testis RNA was reverse transcribed using SuperScript VILO cDNA Synthesis Kit (Invitrogen 11754050). Candidate splice isoforms were characterised by RT-PCR, followed by Qiaxcel capillary gel electrophoresis for quantitation. Percentage Splicing Inclusion (PSI) values

were calculated using the formula: PSI= [concentration of isoform including alternative event/(concentration of isoform including alternative event + concentration of isoform excluding alternative event)] × 100. Primers used for RT-PCR amplification of endogenous splice isoforms are given in Table EV4.

## iCLIP analysis of adult wild type testes

iCLIP experiments were performed in biological triplicate adult mouse testes following the iCLIP protocol as previously described (Best et al, 2014b; Konig et al, 2011). Briefly, germ cells were released by mincing mouse testes, then irradiated with 400 mJ cm$^{-2}$ ultraviolet-C light on ice, lysed and subject to partial RNase digestion. At high RNase concentrations a single radiolabelled RNA protein adduct of ~40 kDa immunoprecipitated with endogenous Tra2β protein (arrowed in Fig. EV3), corresponding to the approximate molecular weight of endogenous Tra2β protein (37 kDa). The iCLIP replicates were analysed using the iMAPS pipeline execution demultiplex and analyse, and PEKA values were used to assess enrichment of 5-mers near to sites of cross-linking (Kuret et al, 2022). The iCLIP data is deposited at GEO (GSE235085).

## *Ptbp2* minigene analysis

A region of mouse *Ptbp2* intron 8 was amplified from mouse genomic DNA using the primers 5′-AAAAAACAATTGcatttgttgtggcaggaaca-3′ and 5′-AAAAAACAATTGcctttgctctcatgtctccc-3′. The PCR product was digested with *Mfe*1, and cloned into the *Mfe*1 site of the pXJ41 vector, within the intron sequence separating the two β-globin exons. The minigene was transfected into HEK293 cells with expression constructs for either GFP or GFP-Tra2β fusion proteins, and splicing monitored as previously described (Best et al, 2014b; Ramond et al, 2023; Grellscheid et al, 2011a) using primers within the β-globin exons of pXJ41, 5′-GCTCCGGATCGATCCTGAGAACT-3′, and 5′-GCTGCAATAAACAAGTTCTGCT-3′, and within the *Ptbp2* alternative exon 5′-GATTACAGACTGGTGTCATG-3′, followed by Qiaxcel capillary gel electrophoresis. HEK293 cells were recently tested and found clear of mycoplasma infection. Western blot analysis of transfected protein expression used infra-red detection of secondary antibodies and the LICOR Odyssey CLx. The primary antibodies used were: anti-TRA2B (Abcam ab31353), anti-PTBP2 (Proteintech 55186-1-AP), anti-Tubulin (Sigma T6793), anti-Actin (Sigma A5441), and anti-GFP (Clontech 632375). Secondary antibodies used were anti-mouse 800CW (Licor 926-32210), anti-mouse 680RD (Licor 926-68070), anti-rabbit 800CW (Licor 926-32211), and anti-rabbit 680RD (Licor 926-68071). Imaging and densitometric quantification were performed using LICOR Image Studio.

## Analysis of splice isoforms in multiple adult mouse tissues

Five adult mice (8 weeks) were sacrificed and dissected to remove tissues. Tissues were homogenised using Qiagen TissueRuptor and RNA extracted with TRIzol reagent (Invitrogen, 15596026) employing manufacturers' protocols. cDNA was synthesised using SuperScript VILO cDNA Synthesis Kit (Invitrogen 11754050). RT-PCR was used to characterise the levels of PE inclusion of the SR family genes, followed by Qiaxcel capillary gel electrophoresis for quantitation. The

poison exon inclusion heatmap was generated by GraphPad Prism using the mean PSI. Primer sequences are given in Table EV5.

## Quantitative RT-PCR (RT-qPCR)

cDNA was synthesised by reverse transcription of total RNA from four biological replicate 12 dpp wild type and *Tra2b-cPEko* whole mouse testes using Superscript VILO cDNA synthesis kit (Invitrogen, 11754–050). RT-qPCR was performed in triplicate on cDNA using Luna Universal qPCR Master Mix (NEB, M3003L) and the QuantStudio 7 Flex Real-Time PCR System (Life Technologies). Target gene expression levels were normalised using the average of three reference genes: *Actb*, *Gapdh* and *Hprt1*. Results were determined using the $\Delta\Delta C_t$ method. Primer sequences and amplicon sizes are given in Table EV6.

## Statistical analyses

Unless otherwise indicated, statistical analyses of pairwise comparisons were carried out using unpaired t tests with Graphpad Prism.

# Data availability

The RNAseq and iCLIP data generated in this study is deposited in the NCBI Gene Expression Omnibus (GEO accession GSE235085). This can be accessed via the URL https://www.ncbi.nlm.nih.gov/geo/query/acc.cgi?acc=GSE235085.

The source data of this paper are collected in the following database record: biostudies:S-SCDT-10_1038-S44318-024-00344-6.

# Peer review information

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

## Acknowledgements

The work in this report was funded by the Biological and Biotechnological Research Council (BBSRC) grants BB/I006923/1, BB/S008039/1 and BB/W002019/1 and the King Fahad Medical City, Ministry of Health, Kingdom of Saudi Arabia. The Mary Lyon Centre at MRC Harwell delivered the C57BL/6J-Tra2b$^{em1H/H}$ mouse strain as part of its commitment to the Genome Editing Mice for Medicine project funded by the Medical Research Council grant MC_UP_2201/2. RNAseq was carried out by Newcastle University Genomics Core Facility. The authors thank Veronika Boczonadi of the Newcastle University BioImaging Unit for her support & assistance in this work.

## Author contributions

**Caroline Dalgliesh**: Investigation; Writing—original draft; Writing—review and editing. **Saad Aldalaqan**: Investigation; Writing—review and editing. **Christian Atallah**: Software; Investigation. **Andrew Best**: Investigation; Writing—review and editing. **Emma Scott**: Formal analysis; Writing—review and editing. **Ingrid Ehrmann**: Investigation; Writing—review and editing. **George Merces**: Formal analysis; Writing—review and editing. **Joel Mannion**: Investigation. **Barbora Badurova**: Investigation. **Raveen Sandher**: Investigation. **Ylva Illing**: Resources; Writing—review and editing. **Brunhilde Wirth**: Resources; Writing—review and editing. **Sara Wells**: Resources; Methodology; Writing—review and editing. **Gemma Codner**: Resources; Methodology; Writing—review and editing. **Lydia Teboul**: Resources; Methodology. **Graham R Smith**: Data curation; Software; Writing—review and editing. **Ann Hedley**: Supervision. **Mary Herbert**: Supervision; Writing—review and editing. **Dirk G de Rooij**: Investigation; Writing—review and editing. **Colin Miles**: Supervision; Writing—review and editing. **Louise N Reynard**: Supervision; Investigation; Writing—review and editing. **David J Elliott**: Conceptualization; Supervision; Funding acquisition; Investigation; Writing—original draft; Project administration; Writing—review and editing.

Source data underlying figure panels in this paper may have individual authorship assigned. Where available, figure panel/source data authorship is listed in the following database record: biostudies:S-SCDT-10_1038-S44318-024-00344-6.

## Disclosure and competing interests statement

The authors declare no competing interests and approve the current version of the manuscript.

# Expanded View Figures

**Figure EV1.   Generation of mice with germ cell-specific deletion of the *Tra2b* PE.**

(**A**) The *Tra2b* PE was flanked with *LoxP* sites. The blue primers bind only to the WT allele as the forward primer sits in the region that is removed and replaced by the insertion of the 5′ *LoxP* site. Thus, they will only amplify a non-floxed or non-recombined allele. The black reverse primer sits on the 3′ *LoxP* cassette and the forward in the floxed region so will only amplify floxed animals. Neither primer pair will amplify in cre-recombined animals. (**B**) Breeding scheme. *Vasa-Cre* was used to excise the *Tra2b* PE. Male *Vasa-Cre* transgenic mice were mated with female *Tra2bPE*^fl/fl^ mice to obtain *Tra2bPE*^fl/+^;*Vasa-cre* mice. We then mated male *Tra2bPE* ^fl/+^;*Vasa-Cre* mice (the floxed allele will be deleted in mature sperm by the *Vasa-Cre* to generate a deletion allele) with female *Tra2bPE*^fl/fl^ mice to generate four possible genotypes. Actual mouse numbers born of each genotype are shown, along with actual and expected frequencies. (**C**) Example of genotyping result by agarose gel electrophoresis. Lane M marker, lanes 1, 2 presence of the *Vasa-Cre* transgene, lanes 3, 4, 5 different *Tra2b* PE alleles as shown in part (**A**). (**D**) Litter sizes of *Tra2b-cPEko* females and wild type females, after crossing with wild type male mice. Individual litter sizes from wild type female mice are shown as black dots, and the mean as a horizontal line. No litters were obtained from *Tra2b-cPEko* female mice (red dot). 3 breeding cages of each cross were maintained until each of the wild type cages had produced a litter.

## (A) Conditional deletion of *Tra2b* poison exon

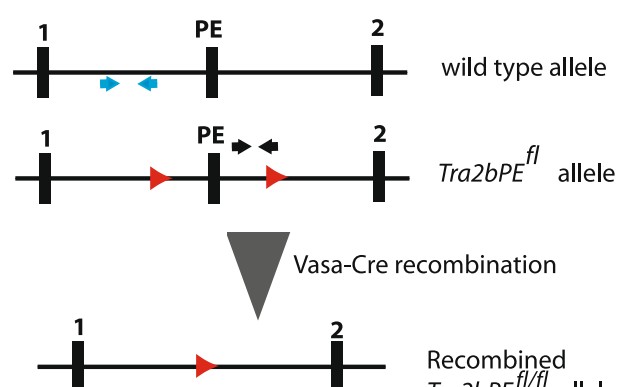

**Key:**
- ➤ PCR primers specific for floxed poison exon
- ➤ PCR primers specific for wild type allele
- ► LoxP site

## (B) Breeding scheme

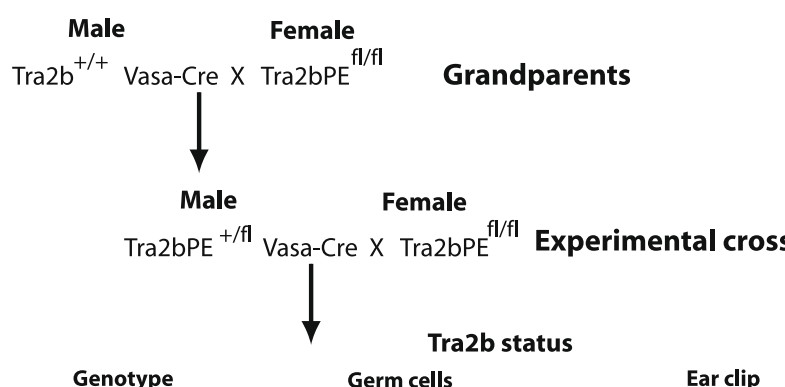

| | | | Tra2b status | | | Mouse numbers | Frequencies | |
|---|---|---|---|---|---|---|---|---|
| | Genotype | | Germ cells | | Ear clip | | Actual | Expected |
| | Tra2bPE$^{+/fl}$ Vasa-Cre | | Tra2bPE$^{+/-}$ **(Tra2b-cPEKO-het)** | | Tra2bPE$^{+/fl}$ | 122 | 0.33 | 0.25 |
| | Tra2bPE$^{-/fl}$ Vasa-Cre | | Tra2bPE$^{-/-}$ **(Tra2b-cPEKO-Hom)** | | Tra2bPE$^{-/fl}$ | 62 | 0.17 | 0.25 |
| | Tra2bPE$^{+/fl}$ | | Tra2bPE$^{+/fl}$ **(wild type)** | | Tra2bPE$^{+/fl}$ | 99 | 0.27 | 0.25 |
| | Tra2bPE$^{-/fl}$ | | Tra2bPE$^{-/fl}$ **(Tra2b-cPEKO-het)** | | Tra2bPE$^{-/fl}$ | 87 | 0.23 | 0.25 |
| | | | | | | **370** | | |

**Experimental genotypes**

## (C) Genotyping analysis (earclip DNA)

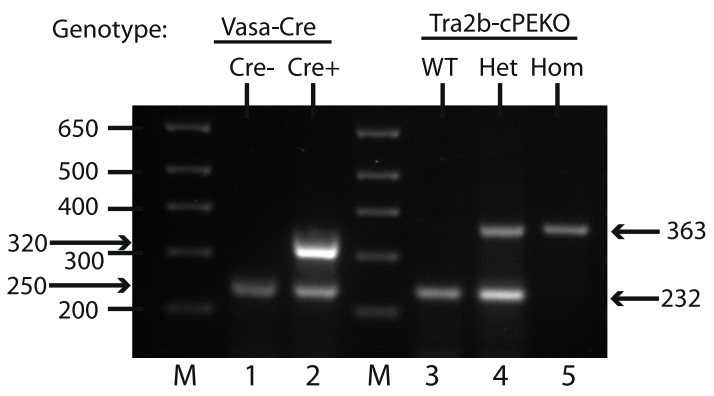

## (D) Female fertility

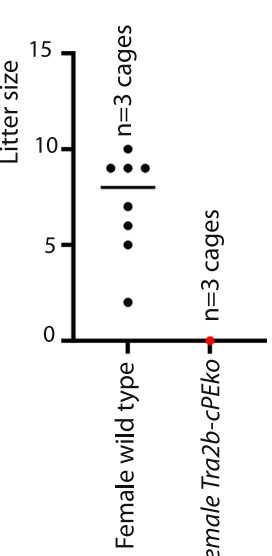

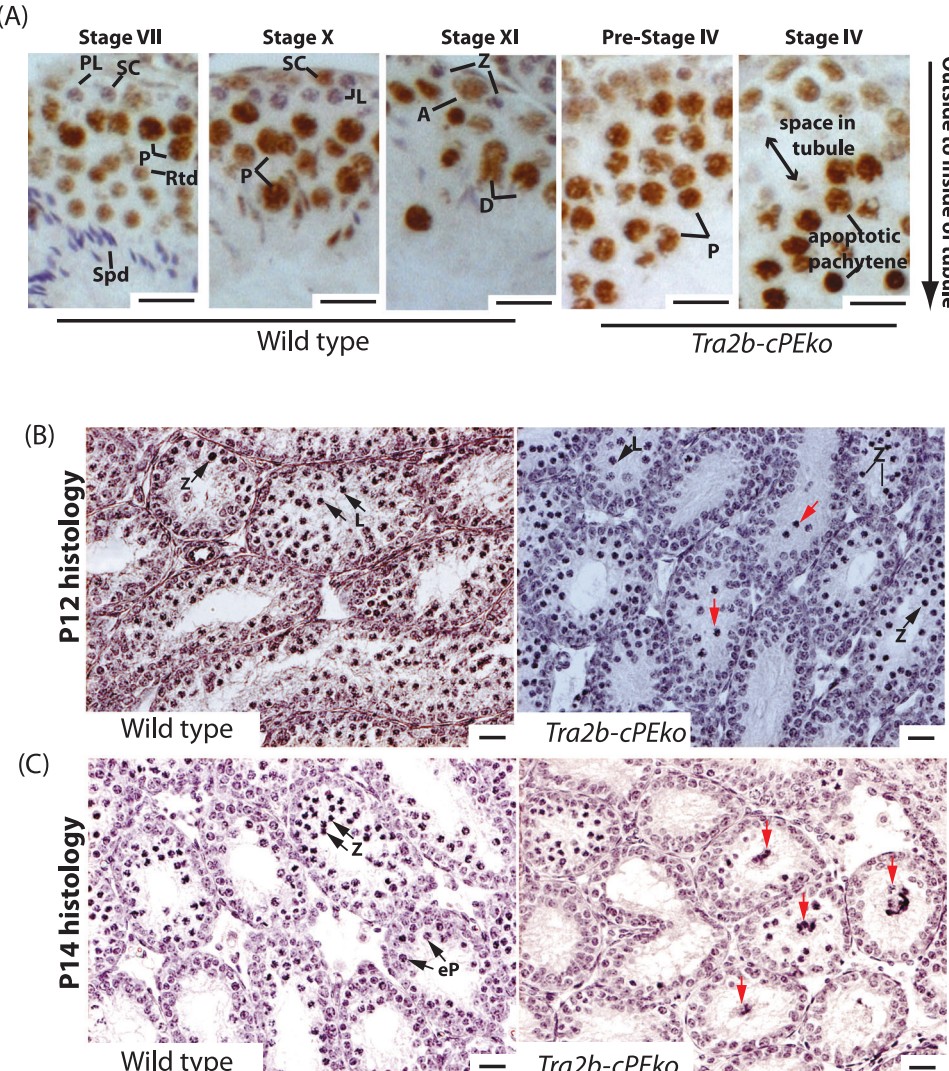

**Figure EV2.   Analysis of the effect of *Tra2b* PE deletion at days P12 and P14 of mouse testis development.**

(A) Micrographs showing immunohistochemical detection of Tra2β protein in sections made from wild type and *Tra2b-cPEko* adult mouse testes (*n* = 2). Segments of seminiferous tubules at different stages are shown to enable all the major stages of postnatal mouse spermatogenesis to be visualised. Abbreviations for cell types shown in Fig. 1B, with the addition of SC (Sertoli Cell). Scale bar = 20 μm. (B, C) Micrographs of Haematoxylin-stained histological sections of (B) P12 mouse testes and (C) P14 mouse testes of different genotypes. Scale bar = 20 μM. Red arrows indicate sloughing germ cells. Scale bars = 20 μm. Sample numbers for each time point, *n* = 4 wild type and *n* = 4 *Tra2b-cPEko*.

**(A)**

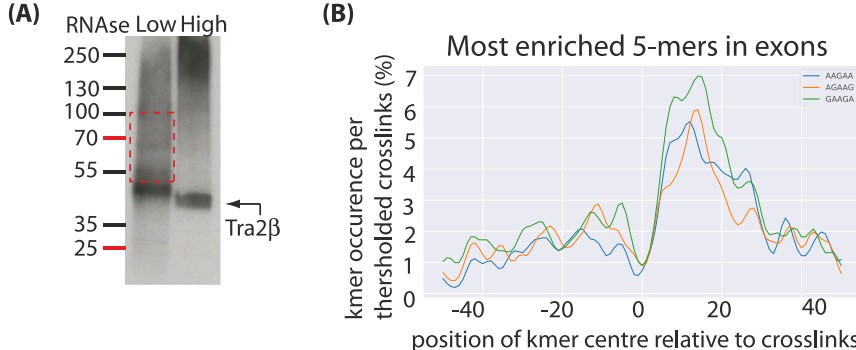

**(B)**

**(C)**

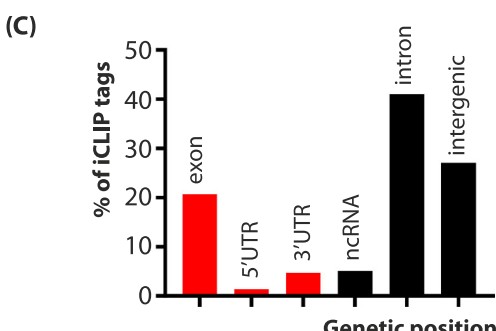

**Figure EV3.   iCLIP Identification of endogenous binding sites for Tra2β in the adult mouse testis.**

(**A**) After cross-linking, endogenous Tra2β protein was immunoprecipitated from wild type mouse testis. Autoradiograph shows 32P-labelled RNA cross-linked to endogenous Tra2β from adult mouse testis during one of three replicate iCLIP experiments. At high RNase concentrations a single radiolabelled RNA-protein adduct of ~40 kDa was detected (arrowed), corresponding to the approximate molecular weight of uncross-linked endogenous Tra2β protein (37 kDa). Tra2β iCLIP tags were recovered at lower RNase concentrations (region used highlighted in red box) in biological triplicate, and these tags were used to map endogenous Tra2β binding sites across the mouse testis transcriptome. (**B**) Analysis of most enriched 5-mers detected close to cross-linking sites within exons (data from one iCLIP replicate shown, similar data were obtained in each of three independent iCLIPs using biological replicate testes). The most frequently occurring pentamers within the iCLIP tags were highly enriched in AGAA nucleotide sequences, exactly corresponding to the known Tra2β binding site (Cléry et al, 2011; Tsuda et al, 2011). (**C**) Genomic distribution of Tra2β binding sites. Average percentage of cross-links from triplicate iCLIP experiments within each genomic region are shown.

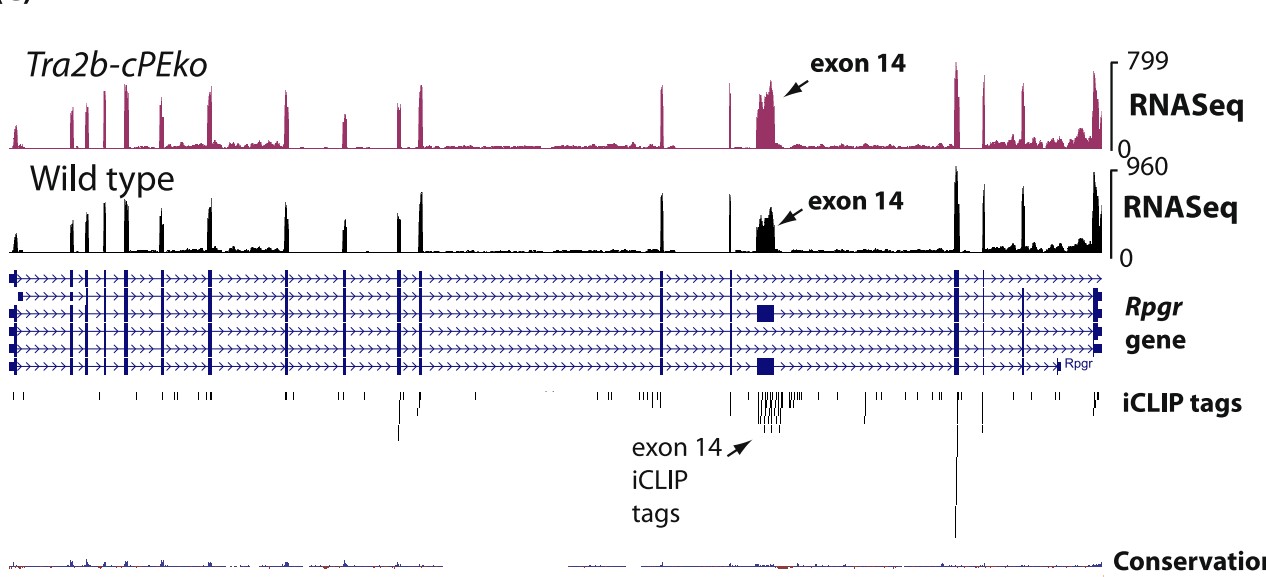

**(A)**

Tra2b-cPEko — RNASeq — 4128 ... 1

exon 6    exon 7

Wild type — RNASeq — 4739 ... 1

exon 6    exon 7

*Ggnbp2* gene

iCLIP tags

exon 6 iCLIP tags

Conservation

**(B)** exon 6 detail

iCLIP tags

consensus binding sites (GAA)

−84,745,203    −84,745,174    −84,745,154    −84,745,135    +84,745,110    −84,745,088
              +84,745,180    −84,745,146                   +84,745,095

E G S S S S V S S E K L S T D K R S S E D H R K D S K C R I I F H Y G P F Q G T A R

**(C)**

Tra2b-cPEko — RNASeq — 799 ... 0

exon 14

Wild type — RNASeq — 960 ... 0

exon 14

*Rpgr* gene

iCLIP tags

exon 14 iCLIP tags

Conservation

◀  **Figure EV4.   Increased inclusion of Tra2β target exons in the *Tra2bcPEko* mouse testis.**

RNAseq reads are merged tracks from testes of 4 P12 mice of different genotypes. Tra2β iCLIP tags are pooled from 3 biological replicate iCLIP experiments using adult wild type mice. (**A**) UCSC mouse genome (mm39) browser screenshot of the *Ggnbp2* gene locus. This screenshot contains the fully expanded iCLIP track, showing an accumulation of iCLIP tags mapping to exon 6. Increased inclusion of *Ggnbp2* exon 6 is detected in *Tra2b-cPEko* mouse testes compared to wild type. (**B**) Detail of exon 6, indicating the positions of experimentally mapped iCLIP tags, and consensus Tra2β protein-RNA binding sites (GAA-containing) sequences. (**C**) UCSC genome browser screenshot showing RNAseq and iCLIP reads aligned to the mouse genome (mm39) at the *Rpgr* locus. Increased inclusion of *Rpgr* exon 14 was detected within the *Tra2b-cPEko* testes compared to wild type.

## (A) Engineering of *Tra2b* exon 4

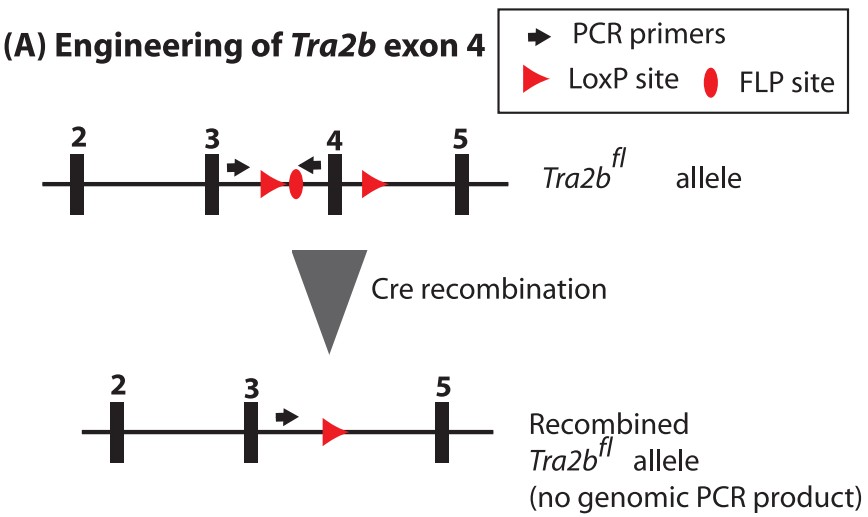

## (B) Breeding scheme

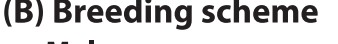

| | Genotype | | *Tra2b* status Germ cells | Ear clip | Mouse numbers | Frequencies Actual | Expected |
|---|---|---|---|---|---|---|---|

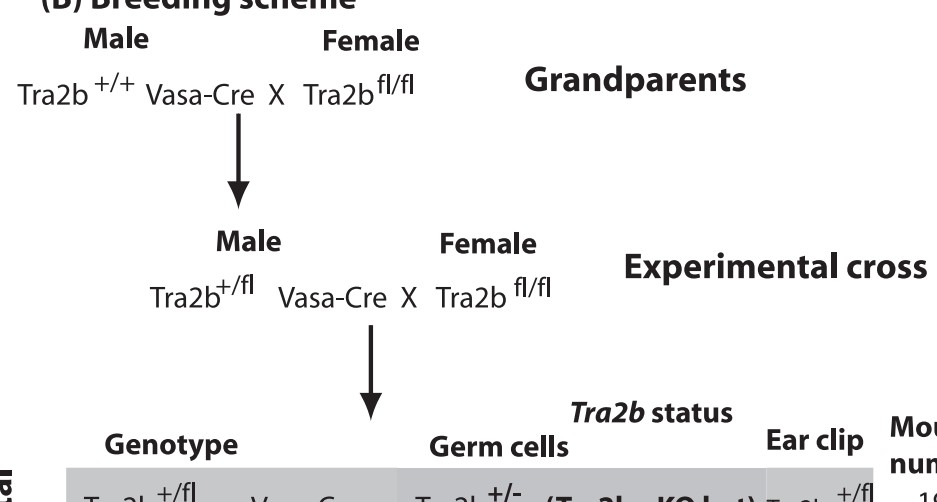

## (C) Genotyping analysis (earclip DNA)

## (D) Female fertility

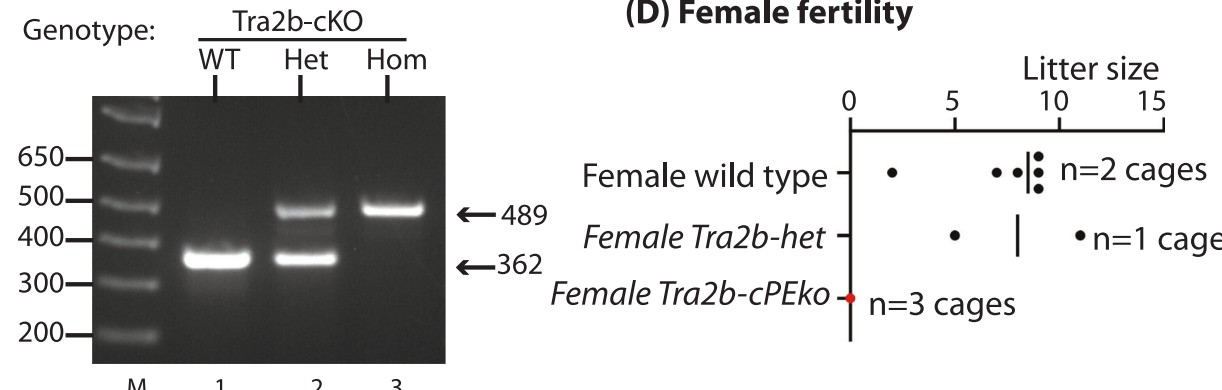

**Figure EV5.  Generation of mice with germ cell-specific deletion of *Tra2b* exon 4.**

(A) An existing conditional *Tra2b^fl* allele in which *Tra2b* exon 4 was flanked with LoxP sites was used to inactivate *Tra2b* gene function. (B) Experimental crosses. Male *Vasa-Cre* transgenic mice were mated with female *Tra2b^{fl/fl}* mice to obtain *Tra2b^{fl/+};Vasa-cre* mice. We then mated male *Tra2b^{fl/+};Vasa-Cre* mice (the floxed allele will be deleted in mature sperm by the *Vasa-Cre* to generate a *Tra2b* knockout allele) with female *Tra2b^{fl/fl}* mice to generate four possible genotypes. Actual mouse numbers born of each genotype are shown, along with actual and expected frequencies. (C) Example of agarose gel electrophoresis genotyping result. Lane M marker, lanes 1, 2, 3 distinguish the different *Tra2b* alleles shown in part (A). (D) Litter sizes of *Tra2b-cko* females, *Tra2b-het* and wild type females, after crossing with wild type male mice. Individual litter sizes from wild type female mice are shown as black dots, and the mean as a horizontal line. No litters were obtained from *Tra2b-cko* female mice (red dot). Breeding cages of each cross were maintained until each of the wild type cages had produced a litter.

