## [Peer Review File · The EMBO Journal]

An ultra-conserved poison exon in the *Tra2b* gene encoding a splicing activator is essential for male fertility and meiotic cell division

Caroline Dalglish, Saad Aldalqaan, Christian Atallah, Andrew Best, Emma Scott, Ingrid Ehrmann, George Merces, Joel Mannion, Barbora Badurova, Raveen Sandher, Ylva Illing, Brunhilde Wirth, Sara Wells, Gemma Codner, Lydia Teboul, Graham Smith, Ann Hedley, Mary Herbert, Dirk de Rooij, Colin Miles, Louise Reynard, and David Elliott

Corresponding author: David Elliott (David.Elliott@ncl.ac.uk)

Review Timeline:

Submission Date:	22nd Nov 23
Editorial Decision:	21st Dec 23
Revision Received:	3rd May 24
Editorial Decision:	16th Jul 24
Authors' Correspondence:	23rd Jul 24
Editor's Correspondence:	25th Jul 24
Revision Received:::	18th Nov 24
Editorial Decision:	29th Nov 24
Revision Received:	4th Dec 24
Accepted:	4th Dec 24

Editor: Ieva Gailite

Transaction Report:

Dear David,

Thank you for submitting your manuscript for consideration by the EMBO Journal. We have now received comments from three reviewers, which are included below for your information.

As you will see from the reports, all reviewers find the study of interest, while also pointing out several important aspects that would need to be addressed in the final version. In particular, they request further characterisation of the Tra2b-cPEko mouse model (referees #1 and #2) and the meiotic defects caused by Tra2 KO (referee #1). Furthermore, referees #2 and #3 indicate that better quantification and statistical analysis should be implemented throughout the manuscript and request more through RNA-seq and iCLIP data analysis and presentation.

Based on the interest expressed in the reports, I would like to invite you to address the issues raised by the referees in a revised manuscript. I think it would be useful to discuss the revision in more detail via email or phone/videoconferencing - please let me know which option you prefer.

We generally allow three months as standard revision time. As a matter of policy, competing manuscripts published during this period will not negatively impact on our assessment of the conceptual advance presented by your study. However, please contact me as soon as possible upon publication of any related work to discuss the appropriate course of action. Should you foresee a problem in meeting this three-month deadline, please contact us to arrange an extension.

When preparing your letter of response to the referees' comments, please bear in mind that this will form part of the Review Process File and will therefore be available online to the community. For more details on our Transparent Editorial Process, please visit our website: <https://www.embopress.org/page/journal/14602075/authorguide#transparentprocess>. Please also see the attached instructions for further guidelines on preparation of the revised manuscript.

Please feel free to contact me if you have any further questions regarding the revision. Thank you for the opportunity to consider your work for publication. I look forward to your revision.

With best wishes,

Ieva

- a point-by-point response to the referees' comments, with a detailed description of the changes made (as a word file).
- a word file of the manuscript text.
- individual production quality figure files (one file per figure)

- a complete author checklist, which you can download from our author guidelines (<https://www.embopress.org/page/journal/14602075/authorguide>).
- Expanded View files (replacing Supplementary Information)
Please see out instructions to authors
<https://www.embopress.org/page/journal/14602075/authorguide#expandedview>

We realize that it is difficult to revise to a specific deadline. In the interest of protecting the conceptual advance provided by the work, we recommend a revision within 3 months (20th Mar 2024). Please discuss the revision progress ahead of this time with the editor if you require more time to complete the revisions.

Referee #1:

This study reported the function of a poison exon in Tra2b in spermatogenesis. Poison exons are ultra-conserved in genes encoding splicing factors. In this study, Tra2b-PE was specifically deleted in mouse germ cells. Tra2b-cPEko males display meiotic arrest and thus are sterile. Further analysis showed that male germ cell fail to proceed through the pachytene stage. RNA-seq analysis of P12 Tra2b-cPEko testes identified differentially expressed genes and differentially spliced genes. The affected exons had high-density of TRA2B CLIP tags. Using a minigene analysis, they demonstrated that cryptic splicing in the Ptbp2 gene is sensitive to TRA2B protein levels. By studying Tra2b cKO mice, they further showed that Tra2b PE is only required for meiotic progression but not for mitotically proliferating male germ cells. The experiments are comprehensive and well executed. The data are of high quality. The conclusions are supported by the data. Overall, the results are very interesting and demonstrate the physiological functions of a poison exon in animals.

I only have minor concerns.

Minor concerns:

1) The meiotic defects in Tra2b-cPEko testes were only characterized at the histological levels. Spermatogenesis failed, apparently at stage IV. I think that it is necessary to characterize meiotic progression of mutant spermatocytes by surface nuclear spread analyses. Such analyses will address the following questions: 1) Is chromosomal synapsis normal in pachytene spermatocytes? SYCP1/SYCP2 or 3 immunofluorescence can be performed. 2) Are there defects in meiotic recombination? This can be analyzed by gammaH2AX and RPA IF on spread nuclei of spermatocytes. Such analyses will provide more molecular details on the meiotic defects seen in the mutant.

2) Vasa-Cre is also active in female germ cells, beginning at E15. Are Tra2b-cPEko females fertile? Are Tra2b-cKO females fertile? A brief description of their phenotypes or lack of phenotypes will provide a complete picture for its role in both male and female germ cells.

Referee #2:

The study is interesting and important to further our understanding of the role of ultraconserved poison exons in vivo, which is a very important topic that needs to be studied. The authors create a unique conditional KO model for the Tra2b-PE in testes to assess its role on germ cell development. They identify defects in spermatogenesis, as well as changes at the expression and splicing level, including in know targets of Tra2b, such as Tra2a or Nasp. They also reveal that Tra2b regulates splicing of Ptbp2. Finally, they show that deletion of the Tra2b gene leads also to defects in spermatogenesis.

Yet, a number of limitations and concerns need to be addressed before these findings are suitable for publication. The biggest issue is that most of the data is presented as representative images without robust quantification and statistical analyses, often without an indication of sample size. There are also critical concerns about the lack of model characterization and validation, as well as the relevance of the RNA-seq data to the model.

MAJOR POINTS:

1) Critical data and information are missing with regards to the characterization and validation of the Tra2b-cPEko mouse model presented in Fig.1 and throughout the paper.

First, there is no genomic data showing whether the PE deletion is actually homozygous within testis, and absent from other tissues. This should be addressed by providing genotyping data from each genotype for multiple animals ($n \geq 5$ per genotype). In addition, one would like to confirm that the relevant cell types within the testis exhibit the Tra2B PE deletion, which should be assessed at the cell type level by DNA FISH and/or single cell approaches along with relevant cell surface markers. Fig. 1C shows PE levels across multiple tissues, including the testis. A very relevant question to this study is exactly which cells within the testis express high levels of these PE?

Second, please provide the frequency of mice obtained from each genotype from the crosses indicated in Fig S1. For Fig.1G-H, Fig. 2A-B. Please show the absence of Tra2B PE-containing transcripts, the levels of the Tra2B coding transcripts in the relevant cell types, as well as changes at the protein level.

2) The biggest issue with this study is that most of the data is presented as representative images without robust quantification and statistical analyses (westerns, and IHC), often without an indication of sample size. This makes it hard to assess the rigor and reproducibility of the data presented. When sample size is shown, there is no justification as to why westerns have 2 samples per condition for some targets, and then 4 samples for the RT-PCR. The lack of consistency in data analyses raises a concern about data rigor and reproducibility. Similarly for the RNA-seq analysis there no indication of how many replicates were used or what the statistical cut offs are.

3) The authors describe an 20% increase in Tra2b transcripts and an 18% increase in Tra2B protein level in their Tra2b-cPEko model as measured by western blot in two animals (Fig. 2E). First, please confirm that this increase in protein is reproducibly detected and statistically significant in multiple animals ($n \geq 5$). Second, these findings are very different from previous work using cellular models (Thomas et al. 2020, Leclair et al. 2020) in which even slight changes in PE-transcript levels lead to 2-10-fold changes in protein levels. This raises the question whether the cells in the Tra2b-cPEko testes are truly homozygotes for the deletion, and if there is a degree of mosaicism in the animals. This is a critical question that needs to be address in order to interpret the data presented across the entire paper.

4) There are several critical questions with regards to the RNA-seq analysis that impact the data interpretation. First, are these experiments carried out using RNA from whole testis? If so, are the rather subtle changes in gene expression or splicing explained by the fact that the RNA-seq is done on whole tissue rather than specifically in germ cell which are the cells that exhibit the Tra2b-cPEko? Second, the authors state a lot of cells die during meiosis in the Tra2b-cPEko, if so is the RNA-seq data even capturing these cells if they aren't there? Is the RNA-seq data just comparing the lack of germ cells to a testis with germ cells rather than measuring the effects of Tra2b-cPEko.

5) The RNA-seq data analysis lacks an unbiased analysis as to the key pathways dysregulated in Tra2b-cPEko animals. Instead, the authors seem to cherry pick a couple of genes associated with male fertility (Fig. 2D). Specifically, these genes do not seem to be affected at the splicing level, raising the question whether these are direct target of Tra2 β and what their relevance in this model is. Similarly, what type of genes are impacted at the splicing level is not discussed (Fig. 3).

6) The authors state that "the splicing targets which had the most activated splicing within the Tra2b-cPEko testes were direct targets for Tra2 β protein binding, and often contained high concentrations of Tra2 β binding sites." based on two example splicing events, one of which is known direct target of Tra2 β . Please conduct a statistical analysis overlapping splicing targets from RNA-seq data and CLIP data.

7) A key finding of the paper is that both the Tra2b-cPEko model which leads to a 20% upregulation of Tra2 β protein, as well as the Tra2b-cko model which leads to loss of Tra2 β protein, are associated with decreased testis size, sperm count, and defects in spermatogenesis. It conceptually hard to understand how both overexpression and knock out of the same protein would lead to a similar phenotype. This needs to be addressed at least in the text.

8) The authors state that "There is a marked temporal difference between when the non-coding Tra2b PE is required during germ cell development compared to the entire Tra2b gene". While the two mouse models use the same Vasa-Cre, to support the claim above the authors need to demonstrate that the timing of the Tra2b-cko and Tra2b-cPEko deletions and their penetrance is the same, as well as the downstream timing of the changes at the protein level.

9) Given that PEs are used for splicing factor cross-regulation (Leclair et al. 2020, Mironov et al. 2023), are there changes detected in other SR proteins PE, besides Tra2a, in Tra2b-cPEko?

10) Given that one of the main effects of the Tra2b-cPEko is impacting the levels of Ptbp2, one would expect at least a subset of the expression and splicing changes would correspond to Ptbp2 targets. This should be analyzed using the RNA-seq in more details.

11) Please explain what the physiological relevance is of using a minigene that takes the cryptic site of PTBP2 completely out of context. Cryptic sites depend on competition with their native splice sites; therefore this experiment is difficult to interpret.

MINOR POINTS

1) Fig. 1A needs more detailed information on the positioning of the CRISPR guides., Do they include the whole the ultraconserved region? And is so how far away are these from it.

2) Please add size of cohorts and ages to Fig 1. For Fig. 1C please specify which animals these come from (including ages and sexes), and indicated cohort size.

3) For Fig.1C-D, 1G-I, 2A-D, 3A, 5B-G, S2A-B, S3A-B, 4C please add quantification of the signal across multiple samples (n{greater than or equal to}5), along with replicates number and statistical analyses.

4) Critical experimental details are missing from the text, figure legends and figures. Please comment on what ages are the fertility studied carried at? What age are the differences in testis size calculated in Fig. 1D-E and do these differences maintain as the animals age? Given the role of testis in controlling male hormones, are there any other phenotypic differences between Tra2b-cPEko animals and controls. Please confirm that the KO is not affecting the function of testosterone producing cells. For figure 1D, please provide images of whole animals, and add whole body size as a control.

5) Please add description of the number of samples used for RNA-seq in Fig. 2D, or how differential expressed genes were defined. The cut off show on Fig. 2D and some examples in Fig. S3 suggest that any difference greater than log2 fold change =0.1 are considered which seems like a very low stringency.

6) Please use the same number of replicates for all assays (IHC, Western blots, RT-PCR) or provide a scientific justification for the differences in sample numbers.

7) There seems to be discrepancy between the Log2-fold change in Tra2a expression shown of Fig. 2D and the description of a -0.89 Log2-fold change in the text. Is the data point correctly shown in Fig. 2D?

8) Please describe the predicted consequences of the splicing events discussed in the text and Fig. 3B, D (impact on protein coding function or known protein domains, introduction of a PTC?).

9) Fig. 4H please show the baseline GFP fusion level in sample 1.

Referee #3:

In their manuscript 'An ultra-conserved poison exon in Tra2b is essential for male fertility and meiotic cell division' Elliott and colleagues present an in vivo model to address the functional importance of an ultra-conserved element in the SR protein Tra2b. This is a very interesting and timely topic, as the in vivo functionality of poison exon in SR proteins is not well understood. These poison exons often represent ultra-conserved elements, strongly pointing to an important function across evolution, but their in vivo relevance remains enigmatic. In the manuscript a germ cell specific knock out of the poison exon of Tra2b is generated, which leads to defects in sperm development and male infertility. This is a very interesting finding, as it associates a defect at a specific developmental stage with an ultra-conserved poison exon, also suggesting that other such exons may have specialized functions in other developmental processes. However, the manuscript would strongly benefit from additional functional or mechanistic insights and some methodological improvements before publication. Specific comments are below, please note that I will mainly focus on alternative splicing and poison exons in this review.

1) It is unclear why the poison exon isoform (Fig. 1c) are so abundantly detected. That would mean that NMD is not/less active in testis. This would be very interesting and should be tested, for example with an NMD reporter. Do components of the NMD pathway show reduced expression in testis?

Fig. 1c is quite central for the manuscript. Please provide the RT-PCR gels that this figure is based upon as supplement. How often were these RT-PCRs done (n=?). Every RT-PCR needs a -RT control that is also shown.

2) It is somewhat surprising that heterozygous animals show a wt phenotype, as one Tra2b gene without the poison exon should still lead to increased Tra2b levels. Please quantify Tra2b protein expression in wt, het and hom testes (e.g. as in Fig. 2a, including het and a quantitative analysis).

3) In all places where the authors say 'splicing inclusion' it should be 'inclusion' (without splicing)

4) Fig. 2d: Please validate all targets considered to be functionally important by RT-qPCR in independent samples. E.g. Maelstrom, Rad51, Mei1, Malat, Ptbp2, Tra2a and Tra2b as they are prominently mentioned in the text.

- 5) The effect in Fig. 2e is very small. Please show all WBs that have been quantified in the supplements. It is unclear how the quantification was done. Are technical duplicates used for the quantification? ('n=3 pairwise comparisons, each independently measured twice')
- 6) Fig. 3: How many targets did the authors try to validate and what is the validation rate? Where independent samples used for the validations in 3b and 3d? If not, this has to be done. Every validation needs a -RT control shown in the figure. As several gels show more than the expected bands, please indicate size markers. For Map7d2 the labeling seems switched, the +exon 7 product should be the larger one.
- 7) What have the regulated exons in common? Do they all show a Tra2b binding site in CLIP data? Are they Short/long/GC/strong or weak splice sites etc.? What renders a particular splice sites hyper-sensitive?
- 8) Is there one splicing event that could be mainly responsible for the phenotype?
- 9) Fig 3a: Why is the dPSI in the Tra2b exon not stronger, as this exon is not present anymore, this should be a black and white splicing switch as there is strong inclusion in wt (Fig. 1C)
- 10) The entire CLIP data is described in a very unprecise way ('...exons most strongly activated in the Tra2b-cPEko P12 mouse testis frequently had high Tra2 β binding site densities.'). What is 'frequently' and what is 'high'. This has to be quantitative and put into more precise numbers. Where else is Tra2b binding? Similarly '...splicing targets which had the most activated splicing within the Tra2b-cPEko testes were direct targets for Tra2 β protein binding, and often contained high concentrations of Tra2 β binding sites' is not precise and not shown. Please provide a quantitative analysis.
- 11) Page 10: 'Increased splicing inclusion of a direct Tra2 β -target exons were also...' should be 'Increased inclusion of a direct Tra2 β -target exon was also...'
- 12) Ptpb2 Intron retention has to be validated by RT-PCR. There are too many background bands in Fig. 4b. Please optimize that RT-PCR and add a -RT control and size marker.
- 13) The Ptpb2 WB is not convincing. Please provide n=3 and quantify. Is the regulation of Ptpb2 associated with the phenotype?
- 14) Fig. 4H does not show a gradual increase of the protein. This WB should be more convincing.
- 15) The minigene system in Fig. 4f could be used to determine what makes an exon hyper responsive. How many Tra2b binding sites are in the wt sequence? Can the authors delete sites and show a gradual reduction in response?
- 16) Figure legend 3c: 'Tra2b PE deletion changes splicing patterns of the Tra2a gene. (E) UCSC genome' delete '(E)'

Point by point response for EMBOJ-2023-116218

This study reported the function of a poison exon in Tra2b in spermatogenesis. Poison exons are ultra-conserved in genes encoding splicing factors. In this study, Tra2b-PE was specifically deleted in mouse germ cells. Tra2b-cPEko males display meiotic arrest and thus are sterile. Further analysis showed that male germ cells fail to proceed through the pachytene stage. RNA-seq analysis of P12 Tra2b-cPEko testes identified differentially expressed genes and differentially spliced genes. The affected exons had high-density of TRA2B CLIP tags. Using a minigene analysis, they demonstrated that cryptic splicing in the Ptbp2 gene is sensitive to TRA2B protein levels. By studying Tra2b cKO mice, they further showed that Tra2b PE is only required for meiotic progression but not for mitotically proliferating male germ cells. The experiments are comprehensive and well executed. The data are of high quality. The conclusions are supported by the data. Overall, the results are very interesting and demonstrate the physiological functions of a poison

Reviewer 1 had only minor concerns:

1) The meiotic defects in Tra2b-cPEko testes were only characterized at the histological levels. Spermatogenesis failed, apparently at stage IV. I think that it is necessary to characterize meiotic progression of mutant spermatocytes by surface nuclear spread analyses. Such analyses will address the following questions: 1) Is chromosomal synapsis normal in pachytene spermatocytes? SYCP1/SYCP2 or 3 immunofluorescence can be performed. 2) Are there defects in meiotic recombination? This can be analyzed by gammaH2AX and RPA IF on spread nuclei of spermatocytes. Such analyses will provide more molecular details on the meiotic defects seen in the mutant.

► **RESPONSE.** This is a good point by the reviewer. However, in practice it would take 2 sequential mouse crosses to generate the experimental line, and to generate enough experimental mice to do this properly we would need 4-6 months, depending on the actual genotype and sex ratios we obtained in each litter (we would need to analyse spreads from at least 3 mice from wild type and Tra2b-cPEko genotypes). Based on the time and cost involved, the editor suggested that although this is a very good suggestion, given it was a minor point and the time and cost involved it should be OK to not do this. We have in fact just submitted a proposal to carry on this project and included these suggested analyses by the reviewer.

2) Vasa-Cre is also active in female germ cells, beginning at E15. Are Tra2b-cPEko females fertile? Are Tra2b-cKO females fertile? A brief description of their phenotypes or lack of phenotypes will provide a complete picture for its role in both male and female germ cells.

► **RESPONSE.** These are also good points. We now include this breeding data for *Tra2b-cPEko* mice (Figure EV1) and *Tra2b-cKO* female mice (Figure EV5).

Referee #2:

The study is interesting and important to further our understanding of the role of ultraconserved poison exons in vivo, which is a very important topic that needs to be studied. The authors create a

unique conditional KO model for the Tra2b-PE in testes to assess its role on germ cell development. They identify defects in spermatogenesis, as well as changes at the expression and splicing level, including in known targets of Tra2b, such as Tra2a or Nasp. They also reveal that Tra2b regulates splicing of Ptbp2. Finally, they show that deletion of the Tra2b gene leads also to defects in spermatogenesis.

Yet, a number of limitations and concerns need to be addressed before these findings are suitable for publication. The biggest issue is that most of the data is presented as representative images without robust quantification and statistical analyses, often without an indication of sample size. There are also critical concerns about the lack of model characterization and validation, as well as the relevance of the RNA-seq data to the model.

MAJOR POINTS:

1) Critical data and information are missing with regards to the characterization and validation of the Tra2b-cPEko mouse model presented in Fig.1 and throughout the paper.

First, there is no genomic data showing whether the PE deletion is actually homozygous within testis, and absent from other tissues. This should be addressed by providing genotyping data from each genotype for multiple animals ($n \geq 5$ per genotype).

► **RESPONSE.** (1) We have clarified and improved this by adding new text. We explain that these mice are mosaics for the poison exon deletion within the testis. This is since *Vasa-Cre* is only expressed within germ cells (active from E15) and not within somatic cells. This means that the Sertoli cells, Leydig cells and myoid cells will still have the poison exon within *Tra2b-PEKO* testes (these somatic cell lineages will be equivalent to the earclip genotype in Figure EV1), and only the germ cells will have the homozygous deletion. This mosaic cell status between the germline and somatic lineages are supported by Tra2 β staining of *Tra2b-cKO* mouse testes in Figure 5G. This showed that Tra2 β expression was removed specifically within spermatogonia and not Sertoli cells of these *Tra2b-cKO* mice.

(2) We have added genotyping data to Figure EV1 and Figure EV5. Genotyping was done with ear clip DNA, and analyses *Vasa-Cre* status and the presence/absence of the conditional alleles, from which we can infer PE status within the germline. We also add numbers of mice genotyped, expected and actual frequencies.

In addition, one would like to confirm that the relevant cell types within the testis exhibit the Tra2B PE deletion, which should be assessed at the cell type level by DNA FISH and/or single cell approaches along with relevant cell surface markers.

► **RESPONSE.** This is a good point. Confirming this, there is already quite a lot of background literature on this topic. Patterns of *Vasa-Cre* mediated recombination have been worked out in detail using very elegant experiments by the Gallardo et al paper we cite in the text. Gallardo et al analysed recombination patterns in mice using a *Rosa26-lacZ* reporter. These experiments showed that within the germline, *Vasa-Cre*-mediated recombination begins at E15, and is >95% efficient in male and female germ cells by birth. The Gallardo paper has been cited over 200 times since 2007, and is the “go to” Cre recombinase for activity in the embryonic germline. In

our manuscript, the IHC-detection of Tra2 β protein within the P1 Tra2b-cKO mice (Figure 5G) support the pattern of recombination described by Gallardo et al. We observed a complete loss of Tra2 β staining in spermatogonia at P1 in the *Tra2b-cKO* mice compared to wild type controls. This shows that Vasa-Cre-mediated recombination is complete by P1 (shown in Figure 5G).

Fig. 1C shows PE levels across multiple tissues, including the testis. A very relevant question to this study is exactly which cells within the testis express high levels of these PE?

► **RESPONSE.** The reviewer makes a good point here. We add a whole new figure to address this (Appendix Figure 1). To make this figure we downloaded adult mouse testis, brain and liver RNAseq data, and testis single cell (Sertoli cell, spermatogonia, spermatocyte, round spermatid) RNAseq data that are available at GEO. Note these publicly available RNAseq data are from adult mouse testis cells rather than the P12 data we have analysed in our study (so also contain cells later in meiosis that have changes in NMD). Alignment of this publicly available mRNAseq data detected high *Tra2b* poison exon inclusion within the adult testis compared to liver and brain (Figure Appendix Figure 1A). Alignment of mRNAseq data from purified adult cell testis types show higher levels of *Tra2b* poison exon containing isoforms within spermatocytes and spermatids compared to spermatogonia and Sertoli cells (Appendix Figure 1B). We refer to this Figure in our manuscript text to bring out these points.

Second, please provide the frequency of mice obtained from each genotype from the crosses indicated in Fig S1. For Fig.1G-H, Fig. 2A-B.

► **RESPONSE.** We add new information to our manuscript. These frequencies are now included within Figure EV1 and EV5.

Please show the absence of Tra2B PE-containing transcripts, the levels of the Tra2B coding transcripts in the relevant cell types, as well as changes at the protein level.

► **RESPONSE.** We (1) add a new Figure (Appendix Figure 6) with new experimental data. These data show that the Tra2b PE is undetectable by RT-PCR within the *Tra2b-cPEko* samples and weakly detected in wild type samples, by RT-PCR in P12 and P14 mouse testis (Appendix Fig.6). At both P12 and P14 the NMD pathway is expected to be operational, which would destabilise poison exon containing transcripts. Please see response above about the levels of *Tra2b* mRNA in different cell types (Appendix Fig.1).

(2) We have reorganised the data in our paper. We discussed these protein levels with the EMBO editor who gave us some extra advice here. Our protein level differences between wild type and *Tra2b-cPEko* are not statistically significant although they show a trend. We have found it difficult to get very convincing protein data although we have tried hard. This could be because both Tra2 β and PTBP2 are themselves developmentally regulated in meiosis, so are likely changing during day P12. This adds extra variability to our data unless we were lucky enough to collect

littermates. There would also be a lag in any splicing change and its corresponding downstream change in protein expression, including the time for protein levels to re-equilibrate. An additional point is we are analysing whole testis by Western, so the protein changes might be much higher within specific cell types within the testis, but diluted out by other cell types. We only get a small amount of protein from a P12 testis, that has also confounded these analyses.

Since our protein data was not statistically significant yet showed an interesting trend, we have now moved it to Appendix Figure 2, where we show data from 6 testes.

2) The biggest issue with this study is that most of the data is presented as representative images without robust quantification and statistical analyses (westerns, and IHC), often without an indication of sample size. This makes it hard to assess the rigor and reproducibility of the data presented. When sample size is shown, there is no justification as to why westerns have 2 samples per condition for some targets, and then 4 samples for the RT-PCR. The lack of consistence in data analyses raises a concern about data rigor and reproducibility. Similarly for the RNA-seq analysis there no indication of how many replicates were used or what the statistical cut offs are.

► **RESPONSE.** We have made changes to our manuscript text. (1) We include information about the numbers of replicates in the figure legends. (2) For the RNAseq analysis we now include the statistical cutoffs both in the main text and the Figure legends. (3) We explain why we chose 4 samples of each genotype in the materials and methods. These were based on ENCODE recommendations, the requirements of the bioinformatics programmes we planned to use, and because we anticipated there could be variation between samples. Although C57Bl6 are inbred we harvested tissue from P12 neonatal mice. P12 is a 24 hour window so we were expecting there could be variability in testes gene expression patterns, based on exactly when the mice were harvested (early or late in this window). We chose to analyse 4 samples for each genotype for the RNAseq (so 8 animals in total for RNAseq). This was to enable statistically robust analysis between individuals. We now include RT-PCR confirmation data from all 8 mice in our supplementary data for the key changes we verified. We also include additional data from P14 testis, that corroborates several of the same splice event switches (in some cases even more strongly, e.g. the *Ptbp2* cryptic splice site). (3) For the Westerns we now show three replicates in the Appendix Figure.

3) The authors describe an 20% increase in Tra2b transcripts and an 18% increase in Tra2B protein level in their Tra2b-cPEko model as measured by western blot in two animals (Fig. 2E). First, please confirm that this increase in protein is reproducibly detected and statistically significant in multiple animals (n{greater than or equal to}5).

► **RESPONSE.** We have toned this result down in our manuscript as this was done using n=3 testes from each genotype (so from 6 animals in total). These protein

preparations were from different animals than the ones that we did RNAseq on, as these latter testes were used for RNA extraction and histology (using the two testes/animal). The change at the level of protein is a tendency rather than a statistically significant result. In contrast the change at the RNA level is statistically very significant (calculated by DESeq2). DESeq2 calculates expression changes based on all other genes, so it is much more reliable than a Western blot binary comparison with tubulin or beta actin.

Second, these findings are very different from previous work using cellular models (Thomas et al. 2020, Leclair et al. 2020) in which even slight changes in PE-transcript levels lead to 2-10-fold changes in protein levels. This raises the question whether the cells in the *Tra2b-cPEko* testes are truly homozygotes for the deletion, and if there is a degree of mosaicism in the animals. This is a critical question that needs to be address in order to interpret the data presented across the entire paper.

► **RESPONSE.** These testes are in fact genetic mosaics between germ and somatic cells – because of the activity of *Vasa-Cre*, only the germ cells have the homozygous PE deletion (see Figure EV1 for experimental genotypes – the earclip genotype would be identical to the somatic cell genotype within the testis). We have added text to make this clearer (see above), along with alignment of publicly available purified cell RNAseq of wild type testis that shows *Tra2b* expression in different testis cell types.

Importantly, although they are mosaics between the germline and somatic cells, we can be sure from the histology that the testes of the *Tra2b-cPEko* mice only contain homozygous PEKO germ cells. If these *Tra2b-cPEko* testes also contained spermatogonial stem cells that were either wild type or heterozygous for the PE deletion, we would expect to see substantial patches of normal, full, spermatogenesis. This is because each normal stem cell would produce large clones of differentiating germ cells at each round of divisions. However, only rare late surviving spermatocytes or round spermatids can be seen in very small numbers (single cells or just a couple).

4) There are several critical questions with regards to the RNA-seq analysis that impact the data interpretation. First, are these experiments carried out using RNA from whole testis? If so, are the rather subtle changes in gene expression or splicing explained by the fact that the RNA-seq is done on whole tissue rather than specifically in germ cell which are the cells that exhibit the *Tra2b-cPEko*? Second, the authors state a lot of cells die during meiosis in the *Tra2b-cPEko*, if so is the RNA-seq data even capturing these cells if they aren't there? Is the RNA-seq data just comparing the lack of germ cells to a testis with germ cells rather than measuring the effects of *Tra2b-cPEko*.

► **RESPONSE.** We have improved our text. We make our text clearer by adding the word “whole” (since these analyses were done with whole testes). P12 was chosen

since the cell type content is equivalent in wild type and *Tra2b-PEko* P12 testis (Figure EV2 and Figure 2B). This means that the RNAseq data is not detecting the lack of germ cells in the *Tra2b-cPEko* testis compared to wild type. We agree the use of whole P12 testis, as opposed to purified cell types, could be a reason for the subtle increase in *Tra2b* expression we observe in *Tra2b-PEko* P12 testis compared to wild type. This increase could in fact be stronger within spermatocytes, but diluted by other cells. According to the classic Bellve et al 1977 JCB paper (cited in our manuscript), 39% of the total testis cell content is Sertoli cell at P12, with the remaining 61% being germ cells.

5) The RNA-seq data analysis lacks an unbiased analysis as to the key pathways dysregulated in *Tra2b-cPEko* animals. Instead, the authors seem to cherry pick a couple of genes associated with male fertility (Fig. 2D). Specifically, these genes do not seem to be affected at the splicing level, raising the question whether these are direct target of *Tra2β* and what their relevance in this model is. Similarly, what type of genes are impacted at the splicing level is not discussed (Fig. 3).

► **RESPONSE.** We have done extra analyses of the recovered genes. We have added this extra analysis as Appendix Figure 4, and refer to it in the text.

6) The authors state that "the splicing targets which had the most activated splicing within the *Tra2b-cPEko* testes were direct targets for *Tra2β* protein binding, and often contained high concentrations of *Tra2β* binding sites." based on two example splicing events, one of which is known direct target of *Tra2β*. Please conduct a statistical analysis overlapping splicing targets from RNA-seq data and CLIP data.

► **RESPONSE.** We have improved the manuscript text to bring out these points. Our Leafcutter analysis identified 157 splicing targets that changed splicing between P12 wild type and *Tra2b-cPEKO* mouse testis. Of these, 23 had a splicing change that was high amplitude when visualised on the IGV genome browser. Of these 23 higher amplitude splicing changes, 18 had *Tra2beta* iCLIP tags, or 78%.

7) A key finding of the paper is that both the *Tra2b-cPEko* model which leads to a 20% upregulation of *Tra2β* protein, as well as the *Tra2b-cko* model which leads to loss of *Tra2β* protein, are associated with decreased testis size, sperm count, and defects in spermatogenesis. It conceptually hard to understand how both overexpression and knock out of the same protein would lead to a similar phenotype. This needs to be addressed at least in the text.

► **RESPONSE.** This is a good point, and we have tried to address this better using the text: "Creation of separate *Tra2b-cPEko* and *Tra2b-cko* mouse models using the same *Vasa-Cre* (active from E15, so causing deletion within embryonic germ cells from which all later stages of germ cell development arise) cause different infertility phenotypes. This indicates both expression and expression control (via PE splicing)

of Tra2 β expression are required for spermatogenesis. Tra2 β expression is needed for mitotically proliferating spermatogonia, with Tra2b-cko cells resulting in a “Sertoli Cell Only” phenotype. Similarly the Tra2b gene is required for mitotic proliferation of cells in culture. In contrast, Tra2b-cPEko spermatogonia still proliferate via mitosis without the *Tra2b* PE, and develop into Tra2b-cPEko spermatocytes that then die resulting in a meiotic arrest phenotype. Since the *Tra2b* PE is also not essential for mitotic proliferation in cultured cells (Thomas et al. 2020; Leclair et al. 2020), its essential role in meiotic prophase was unexpected. However, unlike mitosis, meiotic prophase I is highly transcriptionally active (Aldalaqan et al. 2022). Pachytene cells might be particularly vulnerable to *Tra2b* PE deletion because of their increasing endogenous levels of Tra2 β expression, or might be more sensitive to proper patterns of splice isoforms regulated by normal concentrations of Tra2 β .”

8) The authors state that "There is a marked temporal difference between when the non-coding Tra2b PE is required during germ cell development compared to the entire Tra2b gene". While the two mouse models use the same Vasa-Cre, to support the claim above the authors need to demonstrate that the timing of the Tra2b-cko and Tra2b-cPEko deletions and their penetrance is the same, as well as the downstream timing of the changes at the protein level.

► **RESPONSE.** We have removed the sentence in quotes. Please also see our reply to Point 1 (referee 2) about the extensive literature about *Vasa-Cre*, and our reply to point 3 (referee 2) that if the deletions were not fully or almost fully penetrant we would see clumps of progressing germ cells by histology.

9) Given that PEs are used for splicing factor cross-regulation (Leclair et al. 2020, Mironov et al. 2023), are there changes detected in other SR proteins PE, besides Tra2a, in Tra2b-cPEko?

► **RESPONSE.** According to our DESeq2 analysis (supplementary Dataset 1) no other SR proteins significantly change expression within the *Tra2b-cPEKO* P12 testis transcriptome (with an adjusted p value of less than 0.05).

10) Given that one of the main effects of the Tra2b-cPEko is impacting the levels of Ptbp2, one would expect at least a subset of the expression and splicing changes would correspond to Ptbp2 targets. This should be analyzed using the RNA-seq in more details.

► **RESPONSE.** This is a good point. We did look for known targets from the Licatalosi Cell Reports paper (*Ap2b1*, *Rab28*, *Ap1b1*, *Ap2a1*, *Rapgef1*) but did not see any splicing changes. The timing of the Licatalosi analysis was slightly different (splicing changes analysed at P25). Ptbp2 expression only comes on at late zygote, so P12 represents the earliest stages of *Ptbp2* expression. Licatalosi show

a splice change in the *Ptbp2* cKO at day 13 but not day 11 (in *Rab28*) and we are right in between these two timepoints at P12. Since we did not observe any of these known *Ptbp2* target exons changing we did not add anything extra to the manuscript.

11) Please explain what the physiological relevance is of using a minigene that takes the cryptic site of PTBP2 completely out of context. Cryptic sites depend on competition with their native splice sites; therefore this experiment is difficult to interpret.

► **RESPONSE.** We improved the manuscript by explaining the physiological relevance of the minigene with the following text.

“We tested this model further using a minigene approach. Although a caveat to minigene experiments is that they do not perfectly analyse splicing in the context of normal upstream and downstream splice sites, previous minigene analyses showed that both the *Nasp-T* exon and *Tra2a* poison exon were highly sensitive to Tra2 β protein concentrations in vitro (Grellscheid et al. 2011). We further constructed a minigene that included a large block of *Ptbp2* intron 9 including the cryptic splice flanked by β -globin exons (Fig.4E), and analysed its splicing in HEK293 cells. When co-transfected with GFP, a splice product corresponding to use of the cryptic splice site was not detected, showing this cryptic splice site is not selected at ambient cellular concentrations of Tra2 β (Fig.4F, lane 1). However, RT-PCR analysis clearly detected the cryptic splice product after minigene co-transfection with an expression vector encoding a Tra2 β -GFP fusion protein, with cryptic splice site activation increasing in a dose-dependent fashion (Fig.4F-G, lanes 2-5). In contrast, no cryptic splice site activation was observed after co-transfection of the minigene with an expression vector encoding a version of Tra2 β lacking its RRM (Tra2 β Δ RRM, Fig.4F-G, lane 6). Parallel Western blots showed that the Tra2 β -GFP Δ RRM protein was expressed in HEK293 cells at even slightly higher levels than Tra2 β -GFP (Fig.4F). This data support that the model that cryptic splice site activation in *Ptbp2* responds to increased expression of Tra2 β protein in the *Tra2b-cPEko* testis, for which Tra2 β protein-RNA molecular interactions are important.”

MINOR POINTS

1) Fig. 1A needs more detailed information on the positioning of the CRISPR guides., Do they include the whole the ultraconserved region? And is so how far away are these from it.

► **RESPONSE.** We have improved our text by putting in a sentence to the Methods that the gRNAs are outside of the conserved regions. We give the sequences of the guide RNA in Table EV1. These gRNAs are outside the conserved regions. We have checked that all the information is there so that these can easily be “blatted” by any reader onto the UCSC mouse genome sequence, and then visualised relative to other features of interest including conservation.

2) Please add size of cohorts and ages to Fig 1. For Fig. 1C please specify which animals these come from (including ages and sexes), and indicated cohort size.

► **RESPONSE.** We have improved our text to include this information (in Figures 1 legend and also in methods).

3) For Fig.1C-D, 1G-I, 2A-D, 3A, 5B-G, S2A-B, S3A-B, 4C please add quantification of the signal across multiple samples ($n \geq 5$), along with replicates number and statistical analyses.

► **RESPONSE.** We have added the number of replicates, and statistical analysis where used, to the figure legends.

4) Critical experimental details are missing from the text, figure legends and figures. Please comment on what ages are the fertility studied carried at? What age are the differences in testis size calculated in Fig. 1D-E and do these differences maintain as the animals age? Given the role of testis in controlling male hormones, are there any other phenotypic differences between *Tra2b-cPEko* animals and controls. Please confirm that the KO is not affecting the function of testosterone producing cells. For figure 1D, please provide images of whole animals, and add whole body size as a control.

► **RESPONSE.** We have improved the manuscript by adding this extra information. We add ages within the Figure 1 source data (in excel files), and include the ages of the mice from which the testes came that are shown in the Figure. We have only been able to keep the mice for a limited time and have not done an ageing analysis. We don't notice any other phenotypic effects. *Vasa Cre* is not expressed in Leydig cells. There is no change in expression of the androgen receptor in our RNAseq data suggesting there are no global changes in testosterone. The graph in Figure 1E shows a ratio of testis/body weight. We provide now the actual body and testis weights in the Source Data for Figure 1 (in the original data excel files). While the testes of *Tra2b-cPEko* mice are smaller, the *Tra2bPEko* mice are not distinguishable from wild type or heterozygous littermates without genotyping. All of the data in Figure 1 is adult.

5) Please add description of the number of samples used for RNA-seq in Fig. 2D, or how differential expressed genes were defined. The cut off show on Fig. 2D and some examples in Fig. S3 suggest that any difference greater than \log_2 fold change ≥ 0.1 are considered which seems like a very low stringency.

► **RESPONSE.** We added to the figure legend: “MAplot showing gene expression levels in P12 testis transcriptomes and how they change between testes from wild type and *Tra2b-cPEKO* mice (using RNAseq data from n=4 testes from each genotype, analysed using DESeq2). Genes with an adjusted p value of less than 0.05 are shown as red dots (if upregulated) or blue dots (if downregulated).”

6) Please use the same number of replicates for all assays (IHC, Western blots, RT-PCR) or provide a scientific justification for the differences in sample numbers.

► **RESPONSE.** We have added the numbers of replicates used to our figure legends, and these were designed to be most appropriate scientifically. We are also under an ethical and economic obligation not to generate more animals than we need. Most of our RT-PCRs were based on n=4, as we were confirming the data that was predicted from the RNAseq. We have now added extra data from P14 testis (see below). Our Westerns were done from additional mice. We also used separate mice for the fertility analyses, and designed these to compare fertility of wild type and knockout mice in parallel breeding cages.

7) There seems to be discrepancy between the Log2-fold change in *Tra2a* expression shown of Fig. 2D and the description of a -0.89 Log2-fold change in the text. Is the data point correctly shown in Fig. 2D?

► **RESPONSE.** Figure 2D does show a change of -0.89 (see Y axis).

8) Please describe the predicted consequences of the splicing events discussed in the text and Fig. 3B, D (impact on protein coding function or known protein domains, introduction of a PTC?).

► **RESPONSE.** We improved the text to describe this in now for the key examples we discuss.

9) Fig. 4H please show the baseline GFP fusion level in sample 1.

► **RESPONSE.** We have improved this figure by redoing the experiment and adding this more complete Western. We did this repeat experiment without a gradient, which is hard to get exactly right at low concentrations of transfected *Tra2β* protein, and instead transfected equal levels of protein. Please see also our response to reviewer 3 below.

Referee #3:

In their manuscript 'An ultra-conserved poison exon in Tra2b is essential for male fertility and meiotic cell division' Elliott and colleagues present an in vivo model to address the functional importance of an ultra-conserved element in the SR protein Tra2b. This is a very interesting and timely topic, as the in vivo functionality of poison exon in SR proteins is not well understood. These poison exons often represent ultra-conserved elements, strongly pointing to an important function across evolution, but their in vivo relevance remains enigmatic. In the manuscript a germ cell specific knock out of the poison exon of Tra2b is generated, which leads to defects in sperm development and male infertility. This is a very interesting finding, as it associates a defect at a specific developmental stage with an ultra-conserved poison exon, also suggesting that other such exons may have specialized functions in other developmental processes. However, the manuscript would strongly benefit from additional functional or mechanistic insights and some methodological improvements before publication. Specific comments are below, please note that I will mainly focus on alternative splicing and poison exons in this review.

1) It is unclear why the poison exon isoform (Fig. 1c) are so abundantly detected. That would mean that NMD is not/less active in testis. This would be very interesting and should be tested, for example with an NMD reporter. Do components of the NMD pathway show reduced expression in testis?

► **RESPONSE.** This is a good point. The activity of the NMD pathway within the testis has been studied and we refer to a couple of papers in the following text from the manuscript:

“The high levels of SR protein gene PE-containing isoforms detected within the testis did not necessarily imply functional importance, since they could result from meiotic changes to the NMD environment that may generally stabilise frameshifted transcripts (Shum et al. 2016; Jones and Wilkinson 2017).”

Please note that in particular that these cited papers find that, during meiosis, UPF3B is replaced by UPF3A and antagonises NMD.

Fig. 1c is quite central for the manuscript. Please provide the RT-PCR gels that this figure is based upon as supplement. How often were these RT-PCRs done (n=?). Every RT-PCR needs a -RT control that is also shown.

► **RESPONSE.** We have completely updated Fig. 1C by adding 4 extra mice to the analysis (the original analysis was n=1). We add the underlying RT-PCR gels in our Figure 1 Source Data and add the n number in the legend. We used primers that amplify a product that cross an exon junction, so the primers would not work on genomic DNA. We did experiments to confirm that no signal was obtained from genomic DNA in each tissue sample using a -RT control and *Gapdh/Hprt1* primers. All the original gels are in Figure 1 Source Data.

2) It is somewhat surprising that heterozygous animals show a wt phenotype, as one Tra2b gene

without the poison exon should still lead to increased Tra2b levels. Please quantify Tra2b protein expression in wt, het and hom testes (e.g. as in Fig. 2a, including het and a quantitative analysis).

► **RESPONSE.** Please see above our response to Reviewer 1 where we discuss protein expression. Since the data from homozygote mice was not significant, we have not included data from hets.

3) An all places where the authors say 'splicing inclusion' it should be 'inclusion' (without splicing)

► **Response.** We have improved our manuscript by changing this in the text.

4) Fig. 2d: Please validate all targets considered to be functionally important by RT-qPCR in independent samples. E.g Maelstrom, Rad51, Mei1, Malat, Ptbp2, Tra2a and Tra2b as they are prominently mentioned in the text.

► **RESPONSE.** We have carried out new RT-qPCR analysis to validate the DESeq2 analysis. We chose examples based on how easy it was to design specific intron-spanning primers, so this was not all of them. We add this new data as a new figure panel, and associated text to the manuscript to improve it.

5) The effect in Fig. 2e is very small. Please show all WBs that have been quantified in the supplements. It is unclear how the quantification was done. Are technical duplicates used for the quantification? ('n=3 pairwise comparisons, each independently measured twice')

► **RESPONSE.** We agree with the reviewer here, and now place less emphasis on the Western analysis. Please see above our decision about moving the Westerns analysis to the Appendix Figure. Our main focus is on the RNA analyses for which we have good statistical support.

6) Fig. 3: How many targets did the authors try to validate and what is the validation rate? Where independent samples used for the validations in 3b and 3d? If not, this has to be done. Every validation needs a -RT control shown in the figure. As several gels show more than the expected bands, please indicate size markers. For Map7d2 the labeling seems switched, the +exon 7 product should be the larger one.

► **RESPONSE.** We have improved our text here to add clarity. Bioinformatics analysis predicted 157 splicing targets. We went through two levels of validation. Firstly we visually monitored all predicted splicing patterns by examining our RNAseq on the UCSC and IGV genome browsers. This showed 23 splice changes that were strong enough to see clearly. Of these we further validated the ones shown in the figure by RT-PCR. For the ones that were not clear visual changes, Leafcutter could have been picking up smaller yet statistically significant changes (we could see

some smaller changes at the level of Sashimi plots), so we don't want to necessarily say the stronger changes are the only real ones.

Our new text is: Consistent with this, further bioinformatic analysis using Leafcutter (Li *et al*, 2018) detected a panel of 157 genes with significant splicing differences, defined as having a padjust equal to or less than 0.05 and a Δ PSI (Percentage Splicing Inclusion) of greater than 0.1 between the wild type and *Tra2b-cPEko* P12 testes (**Fig.3A** and **Figure 3 Source Data**). Of these predicted splicing changes, 23 could be easily visualised on a genome browser after alignment of RNAseq reads. We experimentally validated a sample of 4 activated exons by carrying out RT-PCR analyses on the 4 biological replicate wild type and *Tra2b-cPEko* P12 mouse testis RNA samples used for RNAseq (**Fig.3B**). We could also detect most of these same splice changes between 4 independently harvested wild type and *Tra2b-cPEko* P14 mouse testes (**Appendix Figure 5A-E**). These data confirm there are aberrant splicing patterns within the *Tra2b-cPEko* testis, and so confirm that some alternative exons are acutely sensitive to *Tra2b* PE deletion.

Please note that the RNA samples used for validation were the n=4 wild type and n=4 *Tra2b-cPEko* biologically independent samples used for RNAseq (so 8 mice in total). As in previous work from our lab, we have validated splicing changes using the same samples as RNAseq to confirm they are real rather than just trusting our bioinformatics based on short read sequencing. n=4 biologically independent replicates is already quite stringent, and we ignored anything that was not consistent. However, to satisfy the reviewer, we have now also done independent validation on additional biologically independent samples from P14 testis RNA. Most of the splicing changes shown in 3b and 3d were also clearly detected in these additional samples. The exception was *Ggnbp2* exon 11, which we found is in fact also developmentally regulated between day 12 and day 14 even in wild type testes (so the P14 wild type sample looks like the P12 *Tra2b-cPEko* sample, although we don't show this we can if required). We hope these extra samples are sufficient for the reviewer. We add this as new data as Appendix Figure 5, and add the following text to our manuscript:

We experimentally validated a sample of 4 activated exons using RT-PCR on the 4 biological replicate wild type and *Tra2b-cPEko* P12 mouse testis RNA samples used for RNAseq (**Fig.3B**). We could also detect most of these same splice changes between 4 independently harvested wild type and *Tra2b-cPEko* P14 mouse testes (**Fig.S7A-E**).

Just also to further reassure the reviewer about our splicing analyses, we used this splicing analysis to make predictions that we could test experimentally in vitro.

Specifically, our bioinformatics analysis predicted the cryptic splice event in *Ptbp2* intron 8. We used this information to experimentally confirmed that this splice site is Tra2 β -responsive in a minigene experiment. This cryptic event is hardly used in HEK293 cells, but becomes significantly activated after additional Tra2 β co-expression. Thus our original splicing predictions are also further corroborated by additional independent experimentation. Interestingly, the cryptic event in *Ptbp2* intron 8 was a novel event picked up in this study. However, our splicing analysis also identified strong splicing changes within *Tra2a* and *Nasp*, which are already known to respond to Tra2 β levels, and which gives us also extra confidence in our bioinformatics analysis (see Grellscheid et al 2011 *PLOS Genet*, and Best et al 2014 *Nat. Comm.*).

All our analyses used qiaxcel capillary electrophoretograms so we get the sizes from the machine which is calibrated with a size marker. The pattern for *Map7d2* is a bit different. Because the regulated exon is quite long, this experiment uses three primers. This means that the lower band is labelled correctly as exon inclusion, even though this seems a bit counterintuitive. We make this clearer in the legend. All our primers span exon/intron junctions, so genomic DNA cannot be specifically amplified. We add an example –RT control to Fig. S7 for *Ptbp2*, as there were extra bands on this. No signal is detected without RT. The background on this looks quite grey, as the Qiaxcel automatically increases contrast to “look” for a signal.

7) What have the regulated exons in common? Do they all show a Tra2b binding site in CLIP data? Are they Short/long/GC/strong or weak splice sites etc.? What renders a particular splice sites hyper-sensitive?

► **RESPONSE.** This is a good point, and we have looked at this. The responsive exons do have Tra2 β binding sites by iCLIP, and tend to have weak 3' splice sites compared to the flanking exons. However, we don't want to add this to the text as we are not totally confident about how important these features are without testing them (they could also be true of many alternative exons), particularly since our sample size is small. This important question will require future analysis.

8) Is there one splicing event that could be mainly responsible for the phenotype?

► **RESPONSE.** This is an interesting point for further work. Even just deletion of the non-coding Tra2b poison exon is sufficient to cause germ cell death during meiosis.

We have improved the manuscript with the following text:

“A number of the splice changes in *Tra2b-cPEko* testes are within genes important for germ cell development, but whether one in particular causes the phenotype or a combination requires further analysis.”

9) Fig 3a: Why is the dPSI in the Tra2b exon not stronger, as this exon is not present anymore, this should be a black and white splicing switch as there is strong inclusion in wt (Fig. 1C)

► **RESPONSE.** Good point. We were initially expecting this to be a black and white switch but the actual situation is more complex because these mice are mosaics – the somatic cells within the testis still have the poison exon. Also, at P12 NMD is still operating in most cells, so the poison exon containing Tra2b transcript would be unstable. We can though see a clear splicing switch by RT-PCR – both at P12 and P14, and include this in the supplementary data (Appendix Figure 6).

10) The entire CLIP data is described in a very unprecise way ('...exons most strongly activated in the Tra2b-cPEko P12 mouse testis frequently had high Tra2 β binding site densities.'). What is 'frequently' and what is 'high'. This has to be quantitative and put into more precise numbers. Where else is Tra2b binding? Similarly '...splicing targets which had the most activated splicing within the Tra2b-cPEko testes were direct targets for Tra2 β protein binding, and often contained high concentrations of Tra2 β binding sites' is not precise and not shown. Please provide a quantitative analysis.

► **RESPONSE.** We have improved the text by adding the percentage of strong targets that have iCLIP tags, and more analysis of associated splice sites. We also add some more details about the distribution of iCLIP tags as Figure EV3. Our manuscript now has the additional text:

Almost 27% of unique iCLIP tags mapped to exons (internal exons plus 5' and 3' untranslated regions), even though exons only make up 1% of the genome (Fig.EV3C). This iCLIP data also revealed that 18 of the 23 higher amplitude splicing changes (76%) had associated Tra2 β iCLIP tags, indicating they are direct targets for Tra2 β binding within adult mouse testis. These included the *Tra2a* gene PE, that contained multiple iCLIP tags and was also previously known to bind to and be activated by Tra2 β (Grellscheid *et al*, 2011; Best *et al*, 2014b).

11) Page 10: 'Increased splicing inclusion of a direct Tra2 β -target exons were also...' should be 'Increased inclusion of a direct Tra2 β -target exon was also...'

► **RESPONSE.** This is corrected.

12) Ptbp2 Intron retention has to be validated by RT-PCR. There are too many background bands in Fig. 4b. Please optimize that RT-PCR and add a -RT control and size marker.

► **RESPONSE.** We agree this is quite a subtle difference, so we have taken this out of the manuscript.

13) The Ptbp2 WB is not convincing. Please provide n=3 and quantify. Is the regulation of Ptbp2 associated with the phenotype?

► **RESPONSE.** Please see our discussion of protein levels above. We show n=3 and quantify in appendix figure 2. Knockout of Ptbp2 does cause a phenotype during meiosis, but it is only one of several genes that change splicing so it is difficult to know if this change is causative.

14) Fig. 4H does not show a gradual increase of the protein. This WB should be more convincing.

► **RESPONSE.** To make this more convincing we now show a Western with single samples at a higher concentration, and the corresponding RT-PCRs. This is since the original gradual increases of protein concentration are difficult to show at such low transfection concentrations. The important things are there is no selection of the cryptic splice site after transfection with GFP or with the Δ RRM, and no endogenous selection of the cryptic splice site at endogenous concentrations of Tra2 β .

15) The minigene system in Fig. 4f could be used to determine what makes an exon hyper responsive. How many Tra2b binding sites are in the wt sequence? Can the authors delete sites and show a gradual reduction in response?

► **Response** We would like to dissect this further in the future. However, we have already done these kinds of minigene experiment for several exons so we just add some extra text to cover these already published analyses: *NASP-T* (*NASP-T* was originally identified by HITS-CLIP in the mouse testis, and also mutated in this same 2011 PLOS Genet paper, and also identified now to respond to loss of the PE). We have also done similar experiments on human *HIPK3-T*, *CHEK1* exon 4 and a constitutive exon within *ANKRD1*. Each of these minigene experiments show that splicing response to Tra2 β is dependent on binding site numbers. To clarify that binding site numbers impact Tra2 β -responsiveness we now refer to these studies in the text.

16) Figure legend 3c: 'Tra2b PE deletion changes splicing patterns of the Tra2a gene. (E) UCSC genome' delete '(E)'

► **RESPONSE.** This is corrected.

Dear David,

Thank you for submitting your revised manuscript for consideration by The EMBO Journal. I have now received comments from two of the original reviewers, which are included below for your information.

Unfortunately, both reviewers find that some of their points were not addressed satisfactorily, which converge on the evidence of Tra2b poison exon deletion. In particular, reviewer #3 finds that the lack of a clear evidence for increased Tra2b protein levels upon Tra2b poison exon excision affects the conclusiveness of the central claims of the manuscript. I appreciate that you had indicated during the revision consultation that addressing this aspect is not possible due to the lack of material. However, since reviewer #3 makes a convincing argument for the relevance of this data, I am afraid that I cannot offer publication of the manuscript in The EMBO Journal in its current form. However, if you find that you have material, such as tissue sections, available for the DNA FISH or quantitative IF experiments, I would be happy to reconsider the decision.

Alternatively, in the interest of a rapid publication of the study I have discussed your manuscript and referee comments with my colleague Martina Rembold at our sister journal EMBO Reports. I am glad to say that she would be interested in considering your manuscript for publication at their journal with added discussion of the remaining concerns and toning down of the conclusions on causality, i.e., that elevated levels of Tra2b cause the failure of meiosis progression, as outlined by referee #3.

You do not need to revise the manuscript prior to transfer. Once you have initiated the transfer, Martina will send you an invitation to revise, outlining the scope of revision. You can also contact her in advance at m.rembold@emboreports.org to find out more about the transfer and the required revisions. If you find the transfer option of interest, please use the link below to transfer the manuscript:

emboj.msubmit.net/cgi-bin/main.plex?el=A1Ii2BAay4B1mY3X7A9ftd7nOZxKK2QEZeUW10D9Y3wY.

Thank you in any case for the opportunity to consider this manuscript. I am sincerely sorry that I could not communicate more positive news, and I very much hope that you will find the transfer of interest.

With kind regards,

leva

leva Gailite, PhD
Senior Scientific Editor
The EMBO Journal
Meyerhofstrasse 1
D-69117 Heidelberg
Tel: +4962218891309
i.gailite@embojournal.org

Referee #2:

Dagliesh et al. have done an extensive amount of work to address the reviewers' comments. The resulting revised text and figures has been improved in terms of scientific rigor and clarity, but there remain several key points that have not been addressed:

1) Please confirm that the relevant cell types within the testis (germ cells) exhibit the Tra2B PE deletion following Cre activation, which can be assessed by DNA FISH and/or single cell approaches.

2) Figures 2A and B provides key data to support the author's claims. Yet they remains difficult to interpret given that they rely on a couple of representative IHC or IF images without any quantification or statistical analyses.

Referee #3:

In the revised version of the manuscript 'An ultra-conserved poison exon in Tra2b is essential for male fertility and meiotic cell division' Elliott and colleagues have addressed many of my initial comments.

However, one initial concern (also raised by reviewer 2) was the quality and small effect of the Tra2b Western blots in PE ko. The answer to this question that there is no significant difference but only a trend can be seen as a deal breaker for the manuscript.

The model, also prominently stated in the abstract 'Failure to proceed through meiosis was associated with increased Tra2b expression sufficient to drive aberrant Tra2 β protein hyper-responsive splice patterns.' and 'Our data indicate PE splicing control prevents toxic levels of Tra2 β protein incompatible with meiotic prophase.' requires increased Tra2b protein level upon deletion of the poison exon. This is the main point of the manuscript: the poison exon controls the level of Tra2b protein during spermatogenesis and if this goes wrong, spermatogenesis is disturbed.

This model can't be true if poison exon deletion does not increase Tra2b protein. This has to be shown as a convincing main figure, either by WB (sorted cells?) or IHC/IF like in Fig. 2A (which has still to be quantified). In the current version there is not even an RT-qPCR of Tra2b in wt vs PE ko.

If this can't be shown it is somewhat hard to imagine how changes in GE and AS in Figs. 2 and 3 can be controlled by unchanged Tra2b protein. It could in fact also be different cell types that make up the testes at that point that lead to altered AS and GE in the two mouse lines, which would be only indirectly related to Tra2b.

If Tra2b protein expression is not changed, deletion of the poison exon would still do something to prevent proper spermatogenesis, but it is then unclear what the mechanism is (it could be very interesting). If there are doubts, a Southern blot of the mouse model could be a first step to detect potential other off target integrations that could also cause a phenotype.

** As a service to authors, EMBO Press provides authors with the possibility to transfer a manuscript that one journal cannot offer to publish to another EMBO publication or the open access journal Life Science Alliance launched in partnership between EMBO Press, Rockefeller University Press and Cold Spring Harbor Laboratory Press. The full manuscript and if applicable, reviewers' reports, are automatically sent to the receiving journal to allow for fast handling and a prompt decision on your manuscript. For more details of this service, and to transfer your manuscript please click on Link Not Available. **

Dear Ieva, thank you for your message. Sorry for my late reply, I was away last week. We do in fact have sections and Caroline has already carried out the indirect immunofluorescence staining. We have a slot next week on our microscope facility to visualise this (we might have to optimise this with a second staining experiment). So if we could please carry out another round of revision for EMBO Journal to enable us to collect this data we would really be grateful. We are also interested to find out the result of this staining experiment, and apologise to the reviewers for not doing this originally (although we did try and improve the Westerns we also had a lot of work to do on the other requested revisions that were successfully completed).

Best wishes, David.

Dear David,

I am glad to hear that you have some samples still preserved that might help to address the final concerns by the referees. If you can obtain conclusive data from this analysis, I will be happy to reconsider a revised version and run it past the original referees for their re-assessment.

With best wishes,

Ieva

Ieva Gailite, PhD
Senior Scientific Editor
The EMBO Journal
Meyerhofstrasse 1
D-69117 Heidelberg
Tel: [+4962218891309](tel:+4962218891309)
i.gailite@embojournal.org

EMBOJ-2024-119640

This is the second revision of EMBOJ-2023-116218R

Dear Ieva,

I am really pleased to let you know that we have finished the quantitative indirect immunofluorescence experiments as requested by reviewers 2 and 3 of our manuscript, and as we discussed in July. Just to remind you, the particular issue was the lack of clear evidence for upregulation of Tra2beta without the poison exon, and this affected our central conclusions from our manuscript submitted to EMBO Journal. We now add new data to address these concerns. Because of the new panels we have split up our figures a bit to avoid over-crowding, so will now have 6 main figures.

New data:

► 1. The immunofluorescence experiments suggested by the reviewers as a solution to quantifying Tra2beta expression levels have worked out really well. Following our discussion, we have now used quantitative indirect immunofluorescence to monitor nuclear expression of Tra2 β and γ H2AX proteins in testis sections made from 6 animals. Since the wild type and Tra2b-PEKO-HET mice were both fertile with identical testis weights and sperm counts we analysed these together, compared to the 3 Tra2b-cPEKO P12 testes that were infertile and azoospermic. At P12 the all of the cell types are synchronised and also equivalent, not missing any cell types because of the poison exon deletion (compared to P14). We also had good quality blocks from these genotypes we could section for this analysis. These experiments give the very clear answer that Tra2 β protein is statistically significantly upregulated in homozygous mice without the Tra2b poison exon (with a ~27% change at the protein level with a p value less than 0.001), compared to the P12 wild type/heterozygous mice. As a control, in these same samples, γ H2AX protein levels did not change in response to poison exon deletion. The upregulation of Tra2beta we detect at the protein level is somewhat similar to the upregulation we see at the RNA level from RNAseq analysis. We added this data as a new panel 3C in the main figures of our manuscript as requested by the reviewers. We also add the original images of these testis sections used for quantitation. Our indirect immunofluorescence analysis utilised a macro that we have deposited at github (<https://github.com/NCL-ImageAnalysis/An-ultra-conserved-poison-exon-in-Tra2b-is-essential-for-male-fertility-and-meiotic-cell-division>).

► 2. Also as requested by the reviewers, we have also used indirect immunofluorescence to quantify the change in expression in Tra2 β protein between adult leptotene and pachytene cells in the wild type testis. We illustrate this with a Stage 9/10 tubule, and also include the original image of the adult wild type testis entire cross section. This is an entirely new Figure 2. Although not shown, we have done additional quantitation of the entire tubule not shown that shows that cells with higher levels of gammaH2AX (i.e. leptotene and zygotene) have lower Tra2 β expression, that we could add if required. However, since different tubules are not synchronised in the adult testis, we thought it more informative to focus on investigating a tubule with the key cell types of interest in Figure 2.

Although the main issue raised by the reviewers and in our subsequent email discussion was the expression levels of Tra2beta, reviewer 2 asked about confirmation that germ cells were subject to Cre mediated recombination. We have also modified our text to make it very clear that we have carried out immunofluorescence analysis of Tra2beta after exon 4 deletion with the same Vasa-Cre in our C57Bl6 mice. This confirms only germ cells are affected by this Cre recombinase, with deletion being completed within spermatogonia (this is in our last figure, and is exactly consistent with previous published data using this Cre line). We have also updated our manuscript to include recently published papers about poison exons.

You also mentioned that EMBO had identified some formatting corrections, and we have also dealt with these:

1. Please provide up to five keywords. **This is now done, given on p2 of manuscript.**
2. Please define the corresponding author on the manuscript title page. **This is now done**
3. CRediT has replaced the traditional author contributions section because it offers a systematic, machine-readable author contributions format that allows for more effective research assessment. Please remove the Author Contributions from the manuscript and use the free text boxes beneath each contributing author's name in our online submission system to add specific details on the author's contribution. More information is available in our guide to authors. **We consulted EMBO J and are adding this at a later stage when the free boxes are available.**
4. Please rename "Conflict of interests" section into "Disclosure and competing interests statement" (further info: <https://www.embopress.org/page/journal/14602075/authorguide#conflictsofinterest>). **This is now done.**
5. Please remove the legends for the source data for Figure 2 and 3 and for the EV tables from the manuscript text and provide them directly in the corresponding files. **This is now done.**
6. Please correct the heading for the supplementary figure legends to "Expanded View Figure Legends" **This is now done.**
7. Our data editors have flagged the following issues in figure legends that need correcting:
 - Please provide the exact p values in the legends of figures 1e; 3b, d; 4b, g; 5c. **This is now done (we assume this meant the ns values?).**
 - Please indicate the statistical test used for data analysis in the legends of figures 2d; 3a. **This is now done.**
 - Please define the box plot in terms of minima, maxima, centre, bounds of box and whiskers, and percentile in the legend of figure 5c. **This plot is now replaced to show the individual values, mean and SD.**
 - Please provide the scale bar for figures 1h; EV 2a. **This is now done.**
 - Please define the scale bar for figures 5f-g. **This is now done.**

- Please note that the scale bar and its definition are missing for figure 1g. **This is now added.**
 - Please note that in figures EV 2a-b the scale bar unit should be corrected from μM to μm (in the figure legend). **This is now done.**
 - Please define the black arrows in the legend of figure 1g, i.
8. For the dataset reference Soumillon et al, 2013, please add the tag “[DATASET]” in the reference list to distinguish it from the other references. Example:
Hörnberg E, Ylitalo EB, Crnalic S, Antti H, Stattin P, Widmark A, Bergh A, Wikström P (2011) Gene Expression Omnibus GSE29650 (<https://www.ncbi.nlm.nih.gov/geo/query/acc.cgi?acc=GSE29650>).
[DATASET] **This is done.**
9. Papers published in The EMBO Journal are accompanied online by a ‘Synopsis’ to enhance discoverability of the manuscript. It consists of A) a short (1-2 sentences) summary of the findings and their significance, B) 3-4 bullet points highlighting key results and C) a synopsis image that is 550x300-600 pixels large (width x height, jpeg or png format). You can either show a model or key data in the synopsis image. Please note that the image size is rather small and that text needs to be readable at the final size. Please send us this information together with the revised manuscript. **We have added a synopsis.**

We think that with these new data our manuscript is very much strengthened, and thank the reviewers for their input into helping with this. Please let me know if you require anything else. Thanks, David.

Dear David,

Thank you for submitting a revised version of your manuscript and addressing the remaining concerns regarding the missing direct evidence for Tra2beta protein level change upon loss of Tra2b poison exon. I have now looked through your point-by-point response and I find it very reasonable. Therefore, there now remain only a few editorial points that need addressing before I can extend official acceptance of the manuscript:

1. Please check that the funding information is correct and identical both in the manuscript and our online system. Currently, King Fahad Medical City (KFMC) is mentioned in our online system but not in the Acknowledgements section.
2. Please upload the Author Checklist file and check if any information, e.g., the figure numbers, needs to be changed in comparison with the R version.
3. We would like to propose some textual edits in the article title, abstract and synopsis. We have also prepared a short blurb that will accompany the title of your manuscript in our online table of contents. Please take a look at the text below and in the attached text file and let me know if any corrections are needed.

Title:

An ultra-conserved poison exon in the Tra2b gene encoding a splicing activator is essential for male fertility and meiotic cell division

Blurb:

The Tra2 β splicing factor fine-tunes its own expression by promoting the inclusion of a conserved poison exon.

Synopsis:

A non-coding "poison exon" (PE) in the Tra2b gene contains a premature termination codon and reduces Tra2 β protein expression via a homeostatic feedback control pathway: at increased levels, Tra2 β splicing factor binds to the poison exon and promotes its inclusion. This study shows that such regulation is essential for male gametogenesis, explaining the high conservation of this exon across vertebrate genomes.

- SR family splicing factor gene transcripts show high poison exon inclusion in the testis.
- Tra2b poison exon deletion leads to male infertility due to germ cell death at meiotic prophase.
- Loss of Tra2b poison exon leads to Tra2 β accumulation and aberrant Tra2 β -dependent splicing patterns.

Please let me know if you have any questions regarding these remaining points. You can use the link below to finalise the manuscript for acceptance.

With best wishes,

leva

leva Gailite, PhD
Senior Scientific Editor
The EMBO Journal
Meyerhofstrasse 1
D-69117 Heidelberg
Tel: +4962218891309
i.gailite@embojournal.org

We realize that it is difficult to revise to a specific deadline. In the interest of protecting the conceptual advance provided by the work, we recommend a revision within 3 months (27th Feb 2025). Please discuss the revision progress ahead of this time with the editor if you require more time to complete the revisions.

The authors addressed the remaining formatting issues.

Dear David,

Thank you for fixing the remaining minor formatting issues and for approving the final textual edits. I am now pleased to inform you that your manuscript has been accepted for publication in the EMBO Journal.

If you have any questions, please do not hesitate to contact the Editorial Office. Thank you for this contribution to The EMBO Journal and congratulations on a nice study!

Best wishes,

Ieva
